# Optical control of ERK and AKT signaling promotes axon regeneration and functional recovery of PNS and CNS in *Drosophila*

Qin Wang[1,2], Huaxun Fan[3], Feng Li[1,2], Savanna S Skeeters[3], Vishnu V Krishnamurthy[3], Yuanquan Song[1,2]*, Kai Zhang[3,4,5,6]*

[1]Raymond G. Perelman Center for Cellular and Molecular Therapeutics, The Children's Hospital of Philadelphia, Philadelphia, United States; [2]Department of Pathology and Laboratory Medicine, University of Pennsylvania, Philadelphia, United States; [3]Department of Biochemistry, Urbana, United States; [4]Neuroscience Program, Urbana, United States; [5]Center for Biophysics and Quantitative Biology, Urbana, United States; [6]Beckman Institute for Advanced Science and Technology, University of Illinois at Urbana-Champaign, Urbana, United States

**Abstract** Neuroregeneration is a dynamic process synergizing the functional outcomes of multiple signaling circuits. Channelrhodopsin-based optogenetics shows the feasibility of stimulating neural repair but does not pin down specific signaling cascades. Here, we utilized optogenetic systems, optoRaf and optoAKT, to delineate the contribution of the ERK and AKT signaling pathways to neuroregeneration in live *Drosophila* larvae. We showed that optoRaf or optoAKT activation not only enhanced axon regeneration in both regeneration-competent and -incompetent sensory neurons in the peripheral nervous system but also allowed temporal tuning and proper guidance of axon regrowth. Furthermore, optoRaf and optoAKT differ in their signaling kinetics during regeneration, showing a gated versus graded response, respectively. Importantly in the central nervous system, their activation promotes axon regrowth and functional recovery of the thermonociceptive behavior. We conclude that non-neuronal optogenetics targets damaged neurons and signaling subcircuits, providing a novel strategy in the intervention of neural damage with improved precision.

**\*For correspondence:**
songy2@email.chop.edu (YS);
kaizkaiz@illinois.edu (KZ)

**Competing interests:** The authors declare that no competing interests exist.

## Introduction

Inadequate neuroregeneration remains a major roadblock toward functional recovery after nervous system damage such as stroke, spinal cord injury (SCI), and multiple sclerosis. Extracellular factors from oligodendrocyte, astroglial, and fibroblastic sources restrict axon regrowth (*Liu et al., 2006*; *Yiu and He, 2006*; *Liu et al., 2011*; *Lu et al., 2014*; *Schwab and Strittmatter, 2014*) but eliminating these molecules only allows limited sprouting (*Sun and He, 2010*), suggesting a down-regulation of the intrinsic regenerative program in injured neurons (*Sun and He, 2010*; *He and Jin, 2016*). The neurotrophic signaling pathway, which regulates neurogenesis during embryonic development, represents an important intrinsic regenerative machinery (*Ramer et al., 2000*). For instance, elimination of the PTEN phosphatase, an endogenous brake for neurotrophic signaling, yields axonal regeneration (*Park et al., 2008*).

An important feature of the neurotrophin signaling pathway is that the functional outcome depends on signaling kinetics (*Marshall, 1995*) and subcellular localization (*Watson et al., 2001*). Indeed, neural regeneration from damaged neurons is synergistically regulated by multiple signaling

**eLife digest** Most cells have a built-in regeneration signaling program that allows them to divide and repair. But, in the cells of the central nervous system, which are called neurons, this program is ineffective. This is why accidents and illnesses affecting the brain and spinal cord can cause permanent damage. Reactivating regeneration in neurons could help them repair, but it is not easy. Certain small molecules can switch repair signaling programs back on. Unfortunately, these molecules diffuse easily through tissues, spreading around the body and making it hard to target individual damaged cells. This both hampers research into neuronal repair and makes treatments directed at healing damage to the nervous system more likely to have side-effects. It is unclear whether reactivating regeneration signaling in individual neurons is possible.

One way to address this question is to use optogenetics. This technique uses genetic engineering to fuse proteins that are light-sensitive to proteins responsible for relaying signals in the cell. When specific wavelengths of light hit the light-sensitive proteins, the fused signaling proteins switch on, leading to the activation of any proteins they control, for example, those involved in regeneration.

Wang et al. used optogenetic tools to determine if light can help repair neurons in fruit fly larvae. First, a strong laser light was used to damage an individual neuron in a fruit fly larva that had been genetically modified so that blue light would activate the regeneration program in its neurons. Then, Wang et al. illuminated the cell with dim blue light, switching on the regeneration program. Not only did this allow the neuron to repair itself, it also allowed the light to guide its regeneration. By focusing the blue light on the damaged end of the neuron, it was possible to guide the direction of the cell's growth as it regenerated.

Regeneration programs in flies and mammals involve similar signaling proteins, but blue light does not penetrate well into mammalian tissues. This means that further research into LEDs that can be implanted may be necessary before neuronal repair experiments can be performed in mammals. In any case, the ability to focus treatment on individual neurons paves the way for future work into the regeneration of the nervous system, and the combination of light and genetics could reveal more about how repair signals work.

circuits in space and time. However, pharmacological and genetic approaches do not provide sufficient spatial and temporal resolutions in the modulation of signaling outcomes in terminally differentiated neurons in vivo. Thus, the functional link between signaling kinetics and functional recovery of damaged neurons remains unclear. The emerging non-neuronal optogenetic technology uses light to control protein-protein interaction and enables light-mediated signaling modulation in live cells and multicellular organisms (*Zhang and Cui, 2015*; *Khamo et al., 2017*; *Johnson and Toettcher, 2018*; *Leopold et al., 2018*; *Dagliyan and Hahn, 2019*; *Goglia and Toettcher, 2019*). By engineering signaling components with photoactivatable proteins, one can use light to control a number of cellular processes, such as gene transcription (*Motta-Mena et al., 2014*; *Wang et al., 2017*), phase transition (*Shin et al., 2017*; *Dine et al., 2018*), cell motility (*Wu et al., 2009*) and differentiation (*Khamo et al., 2019*), ion flow across membranes (*Kyung et al., 2015*; *Ma et al., 2018*), and metabolism (*Zhao et al., 2018*; *Zhao et al., 2019*), to name a few. We have previously developed optogenetic systems named optoRaf (*Zhang et al., 2014*; *Krishnamurthy et al., 2016*) and optoAKT (*Ong et al., 2016*), which allow for precise control of the Raf/MEK/ERK and AKT signaling pathways, respectively. We demonstrated that timed activation of optoRaf enables functional delineation of ERK activity in mesodermal cell fate determination during *Xenopus laevis* embryonic development (*Krishnamurthy et al., 2016*). However, it remains unclear if spatially localized, optogenetic activation of ERK and AKT activity allows for subcellular control of cellular outcomes.

In this study, we used optoRaf and optoAKT to specifically activate the Raf/MEK/ERK and AKT signaling subcircuits, respectively. We found that both optoRaf and optoAKT activity enhanced axon regeneration in the regeneration-potent class IV da (C4da) and the regeneration-incompetent class III da (C3da) sensory neurons in *Drosophila* larvae, although optoRaf but not optoAKT enhanced dendritic branching. Temporally programmed and spatially restricted light stimulation showed that optoRaf and optoAKT differ in their signaling kinetics during regeneration and that both allow spatially guided axon regrowth. Furthermore, using a thermonociception-based behavioral recovery

assay, we found that optoRaf and optoAKT activation led to effective axon regeneration as well as functional recovery after central nervous system (CNS) injury. We note that most of the previous optogenetic control of neural repair studies were based on channelrhodopsion in *C. elegans* (*Sun et al., 2014*), mouse DRG culture (*Park et al., 2015a*) or motor neuron-schwann cell co-culture (*Hyung et al., 2019*). Another study used blue-light activatable adenylyl cyclase bPAC to stimulate neural repair in mouse refractory axons (*Xiao et al., 2015*). These work highlighted the feasibility of using optogenetics to study neural repair but did not pin down the exact downstream signaling cascade mediating neuronal repair. Additionally, most studies focused on peripheral neurons that are endogenously regenerative. Here, we specifically activated the ERK and AKT signaling pathways and performed a comprehensive study of neural regeneration in both the peripheral nervous system (PNS) and CNS neurons in live *Drosophila*. We envision that features provided by non-neuronal optogenetics, including reversibility, functional delineation, and spatiotemporal control, will lead to a better understanding of the link between signaling kinetics and functional outcome of neurotrophic signaling pathways during neuroregeneration.

## Results

### Light enables reversible activation of the Raf/MEK/ERK and AKT signaling pathways

To reversibly control the Raf/MEK/ERK and AKT signaling pathways, we constructed a single-transcript optogenetic system using the p2A bicistronic construct that co-expresses fusion-proteins with the N-terminus of cryptochrome-interacting basic-helix-loop-helix (CIBN) and the photolyase homology region of cryptochrome 2 (CRY2PHR, abbreviated as CRY2 in this work). Following a similar design of the optimized optoRaf (*Krishnamurthy et al., 2016*), we improved the previous optogenetic AKT system (*Ong et al., 2016*) with two tandom CIBNs (referred to as optoAKT in this work) (*Figure 1—figure supplement 1A*). Consistent with previous studies, the association of CIBN and

CRY2 took about 1 s, and the CIBN-CRY2 complex dissociated in the dark within 10 min (*Kennedy et al., 2010*; *Zhang et al., 2014*). The fusion of Raf or AKT does not affect the association and dissociation kinetics of CIBN and CRY2 and multiple cycles of CRY2-CIBN association and dissociation can be triggered by alternating light-dark treatment (*Figure 1—figure supplement 1B–1D*, *Videos 1* and *3*). Activation of optoRaf and optoAKT resulted in nuclear translocation of ERK-EGFP (*Figure 1A*, *Video 2*) and nuclear export of FOXO3-EGFP (*Figure 1B*, *Video 4*) resolved by live-cell fluorescence imaging, indicating activation of the ERK and AKT signaling pathways, respectively.

Western blot analysis on pERK (activated by optoRaf) in HEK293T cells showed that pERK activity (*Figure 1C*) increased within 10 min blue light stimulation and returned to the basal level 30 min after the blue light was shut off (*Figure 1D*). There was a slight decrease in pERK activity upon optoRaf activation for over 10 min, likely due to negative feedback, which has been consistently observed in previous studies (*Zhou et al., 2017*). On the other hand, continuous light illumination maintained a sustained activation of pCRY2-mCh-AKT within an onset of 10 min (*Figure 1E*). The inactivation kinetics of pAKT was 30 min, similar to that of pERK

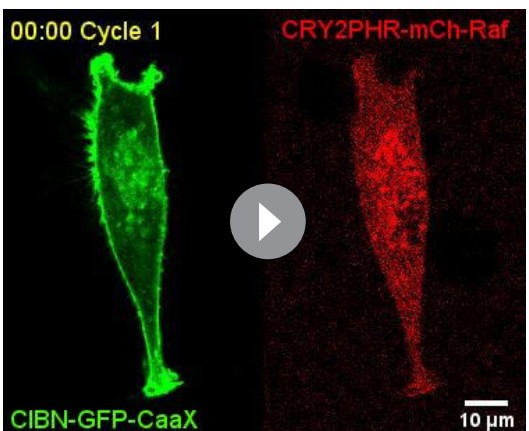

**Video 1.** Reversible optogenetic stimulation of Raf membrane recruitment with optoRaf resolved by live-cell imaging in BHK21 cells. Cells were cotransfected with CIBN-EGFP-CaaX and CRY2-mCh-Raf1, and recovered overnight before imaging. Blue and green light (exposure time 200 ms) were applied every 2 s until the fluorescence intensity of mCherry on the plasma membrane does not change. Cells were left on the microscope in the dark from 30 min to allow for membrane dissociation of CRY2-mCh-Raf. In the next cycle, the same light pattern was repeated and membrane recruitment of CRY2-mCh-Raf was recorded.
https://elifesciences.org/articles/57395#video1

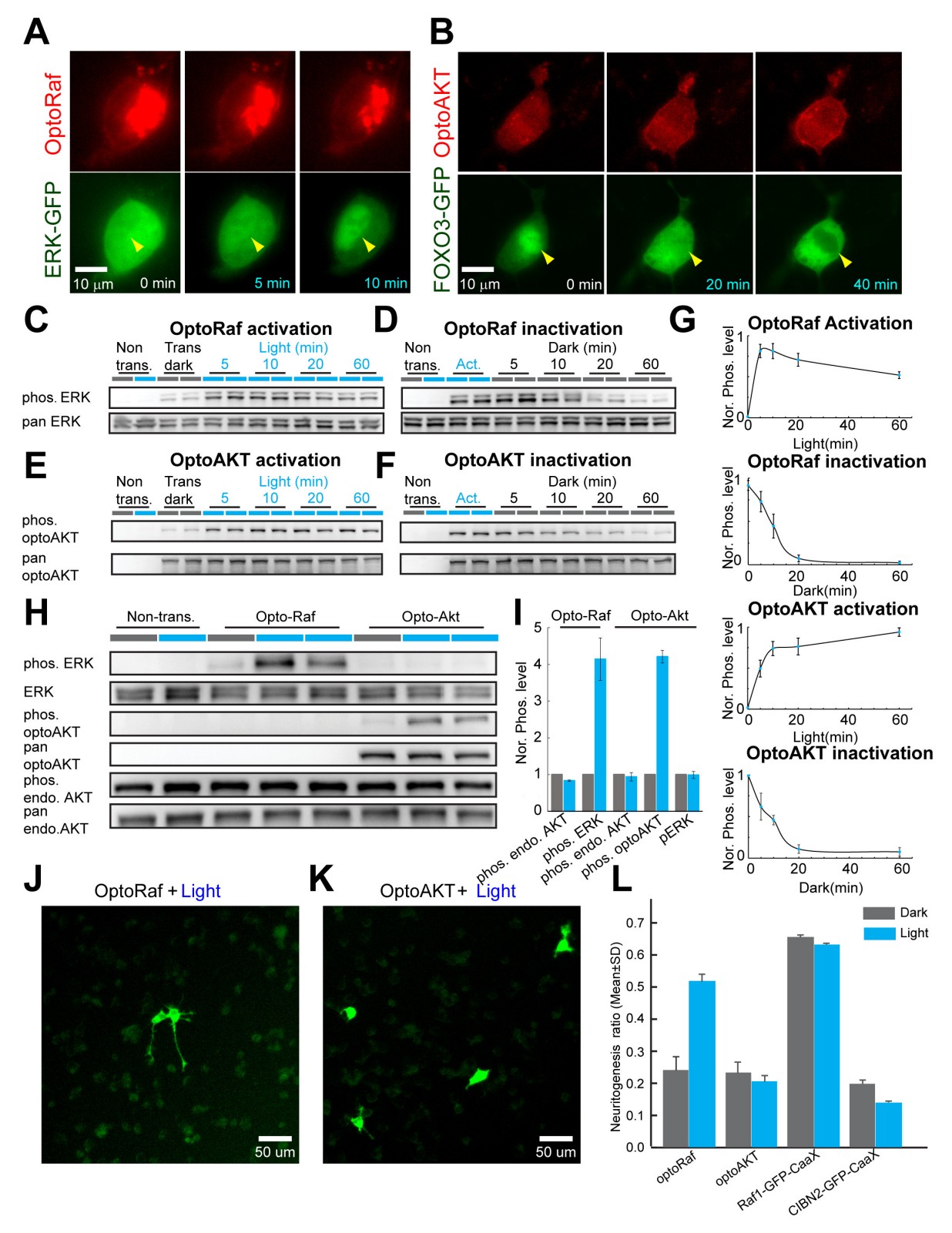

**Figure 1.** OptoRaf and optoAKT specifically activate the ERK and AKT subcircuits, respectively. (**A**) Activation of optoRaf benchmarked with ERK2-EGFP nuclear translocation. (**B**) Activation of optoAKT benchmarked with FOXO3-EGFP nuclear export. Scale bars = 10 μm. (**C**) Western blot analysis of the pERK and ERK activities in response to time-stamped activation of optoRaf. Blue light (0.5 mW/cm$^2$) was applied for 5, 10, 20, and 60 min to HEK293T cells transfected with optoRaf. Non-transfected cells or optoRaf-transfected cells (dark) were used as negative controls. (**D**) Inactivation of the pERK

*Figure 1 continued on next page*

*Figure 1 continued*

activity after blue light was shut off. (**E**) Western blot analysis of the pAKT (S473) and AKT activities in response to time-stamped activation of optoAKT. Cells were treated with identical illumination scheme in (**C**). (**F**) Inactivation of the pAKT activity after blue light was shut off. (**G**) Plots of normalized pERK and pAKT activity upon optoRaf and optoAKT activation, respectively (maximum activation was defined as 1). Both optoRaf and optoAKT show rapid (less than 5 min) and reversible activation patterns (*N* = 3). (**H**) OptoRaf and optoAKT do not show cross activity at the level of ERK and AKT. Cells were exposed to blue light (0.5 mW/cm$^2$) for 10 min before lysis. (**I**) Quantification of the phosphorylated protein level, phosphorylation level was normalized to non-transfected group(*N* = 3). (**J, K**) PC12 cells transfected with either optoRaf (**J**) or optoAKT (**K**) were treated by blue light for 24 hr (0.2 mW/cm$^2$). Scale bars = 50 μm. (**I**) Quantification of the neuritogenesis ratio of PC12 cells transfected with optoRaf or optoAKT. A membrane-targeted Raf (Raf1-EGFP-CaaX) causes constitutive neuritogenesis independent of light treatment, whereas the no-Raf (CIBN2-EGFP-CaaX) control does not increase the neuritogenesis ratio under light or dark treatment. See also *Figure 1—figure supplement 1*.

The online version of this article includes the following source data and figure supplement(s) for figure 1:

**Source data 1.** OptoRaf and optoAKT specifically activate the ERK and AKT subcircuits, respectively.

**Figure supplement 1.** Design and live cell imaging for optoRaf and optoAKT in mammalian cell cultures.

**Figure supplement 1—source data 1.** Design and live cell imaging for optoRaf and optoAKT in mammalian cell cultures.

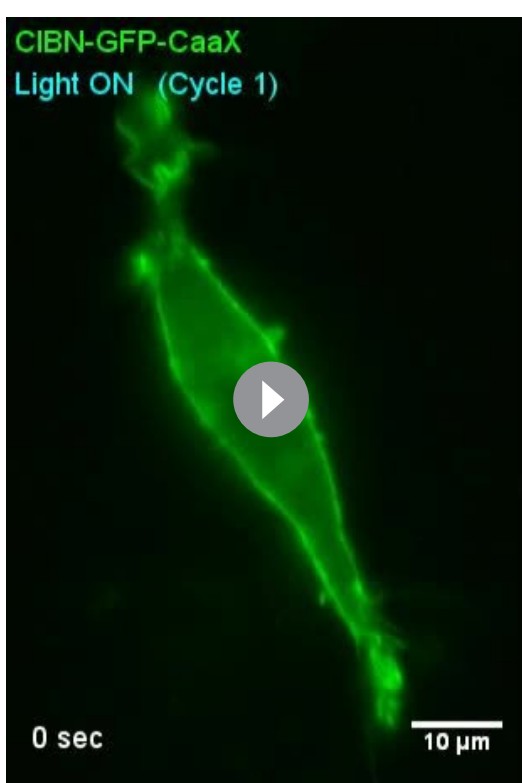

**Video 2.** Reversible optogenetic stimulation of AKT membrane recruitment with optoAKT resolved by live-cell imaging in BHK21 cells. Cells were cotransfected with CIBN-EGFP-CaaX and CRY2-mCh-AKT, and recovered overnight before imaging. Blue and green light (exposure time 200 ms) were applied every 10 s until the fluorescence intensity of mCherry on the plasma membrane does not change. Cells were left on the microscope in the dark from 30 min to allow for membrane dissociation of CRY2-mCh-AKT. In the next cycle, the same light pattern was repeated and membrane recruitment of CRY2-mCh-AKT was recorded.
https://elifesciences.org/articles/57395#video2

(*Figure 1F and G*). Note we use only the phosphorylated and total forms of CRY2-mCh-AKT to quantify the light response of optoAKT because the endogenous AKT does not respond to light.

## optoRaf and optoAKT do not show crosstalk activity at the pERK and pAKT level

Binding of neurotrophins to their receptor activates multiple downstream signaling subcircuits, including the Raf/MEK/ERK and AKT pathways. Delineation of signaling outcomes of individual subcircuits remains difficult with pharmacological assays given the unpredictable off-targets of small-molecule drugs. We hypothesized that optoRaf and optoAKT could delineate signaling outcomes because they bypass ligand binding and activate the intracellular signaling pathway. To test this hypothesis, we probed phosphorylated proteins, including pERK and pAKT with WB analysis in response to light-mediated activation of optoRaf and optoAKT. Results show that optoRaf activation does not increase endogenous pAKT (*Figure 1H and I*). Similarly, optoAKT activation does not increase pERK or endogenous pAKT (*Figure 1H and I*). Thus, at the level of ERK and AKT, optoRaf and optoAKT do not show crosstalk activity in mammalian cells.

## Activation of optoRaf and optoAKT requires upstream signaling molecules

Although activation of both optoRaf and optoAKT bypasses ligand-receptor binding, it remains unclear if other upstream signaling molecules are required to activate optoRaf and optoAKT. Endogenous Raf1 activation requires its membrane translocation mediated by the GTP-bound form of Ras, followed by phosphorylation at several residues, including Ser338, which is located

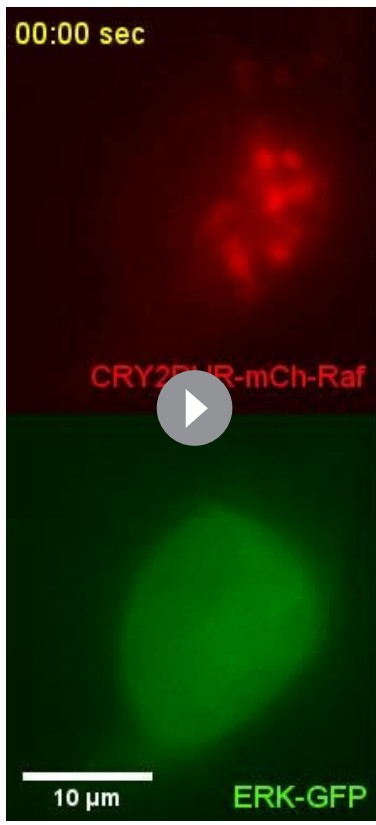

**Video 3.** Optogenetic activation of optoRaf causes nuclear translocation of ERK2-EGFP in BHK21 cells. Cells were transfected with optoRaf (CIBN-CaaX and CRY2-mCh-Raf1) and ERK2-EGFP, and recovered overnight before imaging. Blue and green light (exposure time 200 ms) were applied every 10 s. Nuclear translocation of ERK2-EGFP was recorded. https://elifesciences.org/articles/57395#video3

in the junction region between the regulator domain and the kinase domain (*Mason et al., 1999*). Replacement of Ser338 with alanine abolishes Raf activation (*Xiang et al., 2002*; *Goetz et al., 2003*). Note that phosphorylation of Ser338 itself does not activate Raf but is a prerequisite regulatory event for Raf activation (*Diaz et al., 1997*), likely leading to Raf dimerization (*Takahashi et al., 2017*). To determine if Ser338 phosphorylation is involved in optoRaf activation, we probed the phosphorylation state of CRY2-mCh-Raf upon blue light stimulation and found that indeed Ser338 is significantly phosphorylated upon blue light stimulation (*Figure 1—figure supplement 1E*, top panel, lane 5–8). We then constructed the S338A mutant for optoRaf and confirmed that no phosphorylation occurs on S338 in optoRaf S338A. Importantly, optoRafS338A mutant significantly reduced the blue-light-mediated pERK activation (*Figure 1—figure supplement 1E*, third panel). This result suggests that optoRaf activation through membrane translocation requires upstream kinases.

Similarly, for optoAKT, it remains unclear if its activation requires upstream PI3K signaling. Full activation of AKT requires phosphorylation on both T308 in the activation loop of the catalytic protein kinase core and S473 in a C-terminal hydrophobic motif (*Manning and Toker, 2017*). PH-domain-containing kinases such as PDK1 are essential for AKT activation by phosphorylating AKT on T308, whereas the mechanistic target of rapamycin (mTOR) complex 2 (mTORC2) phosphorylates AKT on S473. In addition to verifying that phosphorylation of S473 occurs during optoAKT activation (*Figure 1E* and *Figure 1—figure supplement 1F*), we probed pT308 for optoAKT upon blue light stimulation (*Figure 1—figure supplement 1F*). We found that light stimulation indeed enhances the level of pT308 in optoAKT, indicating that upstream kinases (e.g. PDK1) are involved in the activation of optoAKT upon membrane translocation of CRY2-mCh-AKT.

## Activation of optoRaf but not optoAKT enhances PC12 cell neuritogenesis

We verified that the activation of optoRaf enhances PC12 cell neuritogenesis, which is consistent with previous studies (*Zhang et al., 2014*; *Krishnamurthy et al., 2016*). The neuritogenesis ratio is defined as the ratio between the number of transfected cells with at least one neurite longer than the size of the cell body and the total number of transfected cells. Twenty-four hours of blue light stimulation (0.2 mW/cm$^2$) increased the neuritogenesis ratio from the basal level (0.24 ± 0.04) to 0.52 ± 0.03 (*Figure 1J and L*). Light-mediated activation of optoAKT, on the other hand, did not increase the neuritogenesis ratio (0.23 ± 0.04 in the dark versus 0.20 ± 0.02 under light) (*Figure 1K and L*). A membrane-targeted Raf1 (Raf1-EGFP-CaaX) was used as a positive control, which caused significant neurite outgrowth independent of light treatment (0.65 ± 0.01 in the dark versus 0.63 ± 0.01 under light). Expression of CIBN2-EGFP-CaaX (without CRY2-Raf1), a negative control, did not increase PC12 neurite outgrowth either in the dark (0.20 ± 0.02) or under light (0.14 ± 0.01) (*Figure 1L*).

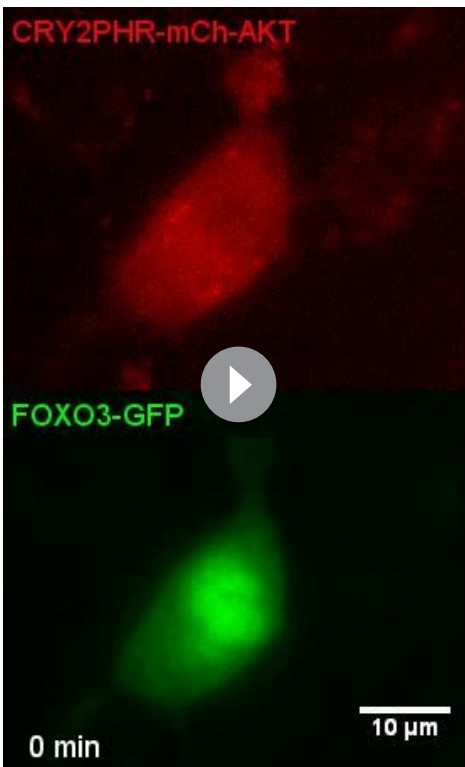

**Video 4.** Optogenetic activation of optoAKT causes retreatment of FOXO3-EGFP from the nucleus into the cytoplasm in BHK21 cells. Cells were transfected with optoAKT (CIBN- CaaX and CRY2-mCh-AKT) and FOXO3-EGFP, and recovered overnight before imaging. Blue and green light (exposure time 200 ms) were applied every 1 min. Nuclear export of FOXO3-EGFP was recorded.
https://elifesciences.org/articles/57395#video4

## Activation of optoRaf but not optoAKT increases sensory neuron dendrite branching in fly larvae

To determine the efficacy of the optogenetic tools in vivo, we generated transgenic flies with inducible expression of optoRaf (*UAS-optoRaf*) and optoAKT (*UAS-optoAKT*). We induced the expression of the transgenes in a type of fly sensory neurons, the dendritic arborization (da) neurons, which have been used extensively to study dendrite morphogenesis and remolding (*Gao et al., 1999*; *Grueber et al., 2002*; *Sugimura et al., 2003*; *Kuo et al., 2005*; *Williams and Truman, 2005*; *Kuo et al., 2006*; *Williams et al., 2006*; *Parrish et al., 2007*). Using the *pickpocket (ppk)-Gal4*, we specifically expressed optoRaf in the class IV da (C4da) neurons, to test whether light stimulation would activate the Raf/MEK/ERK pathway. At 72 hr after egg laying (h AEL), wild-type (WT) and optoRaf-expressing larvae were anesthetized with ether and subjected to whole-field continuous blue light for 5, 10 and 15 min, while as a control, another transgenic group was incubated in the dark (0 min). The larval body walls were then dissected and immunostained with the pERK1/2 antibody, as a readout of the Raf/MEK/ERK pathway activation. We found that 5 min light stimulation was sufficient to significantly increase the pERK signal in the cell body of C4da neurons in optoRaf-expressing larvae, while 15 min illumination enhanced pERK activation and induced ERK translocation into the nucleus (*Figure 2—figure supplement 1A and B*). Compared with the optoRaf-expressing larvae incubated in the dark, 15 min light illumination resulted in a more than twofold increase of pERK fluorescence intensity (*Figure 2—figure supplement 1B*). Similarly, in C4da neurons expressing optoAKT, the 10- and 15 min blue light stimulation significantly increased the fluorescence intensity of phospho-p70 ribosomal S6 kinase (phospho-p70$^{S6K}$) (*Figure 2—figure supplement 1C and D*), which functions downstream of AKT (*Lizcano et al., 2003*; *Miron et al., 2003*). These results collectively demonstrate that optoRaf and optoAKT were robustly expressed in flies and blue light is sufficient to activate the optogenetic effectors in vivo.

The phosphorylation of ERK/p70$^{S6K}$ in response to blue light was only observed in C4da neurons but not in other classes of da neurons or epithelial cells (*Figure 2—figure supplement 2*), proving they are triggered by optoRaf/optoAKT, which were only expressed in C4da neurons under the control of *ppk-Gal4*. Furthermore, we found ERK was not activated in optoAKT-expressing neurons (*Figure 2A*, right-most panel), nor was phospho-p70$^{S6K}$ in the optoRaf-expressing larvae (*Figure 2— figure supplement 3*, right-most panel), confirming that there is no crosstalk between these two systems, at least at the node of pERK and p70$^{S6K}$. We also examined the inactivation kinetics of ERK/ phospho-p70$^{S6K}$ after blue light was shut off (*Figure 2A–D*). The pERK (*Figure 2C*) and pAKT (*Figure 2D*) activity started to decrease as the light was shut off, although the decay rate of pERK decays appears slower than pAKT. Compared with the transgenic larvae kept in the dark, there was no significant difference in phospho-p70$^{S6K}$ intensity at 15 min after blue light was turned off (*Figure 2D*). In contrast, a 15-min off time reduces pERK activity, but the level remains higher than the basal level. When the off-time was increased to 45 min, there is still a slightly higher pERK

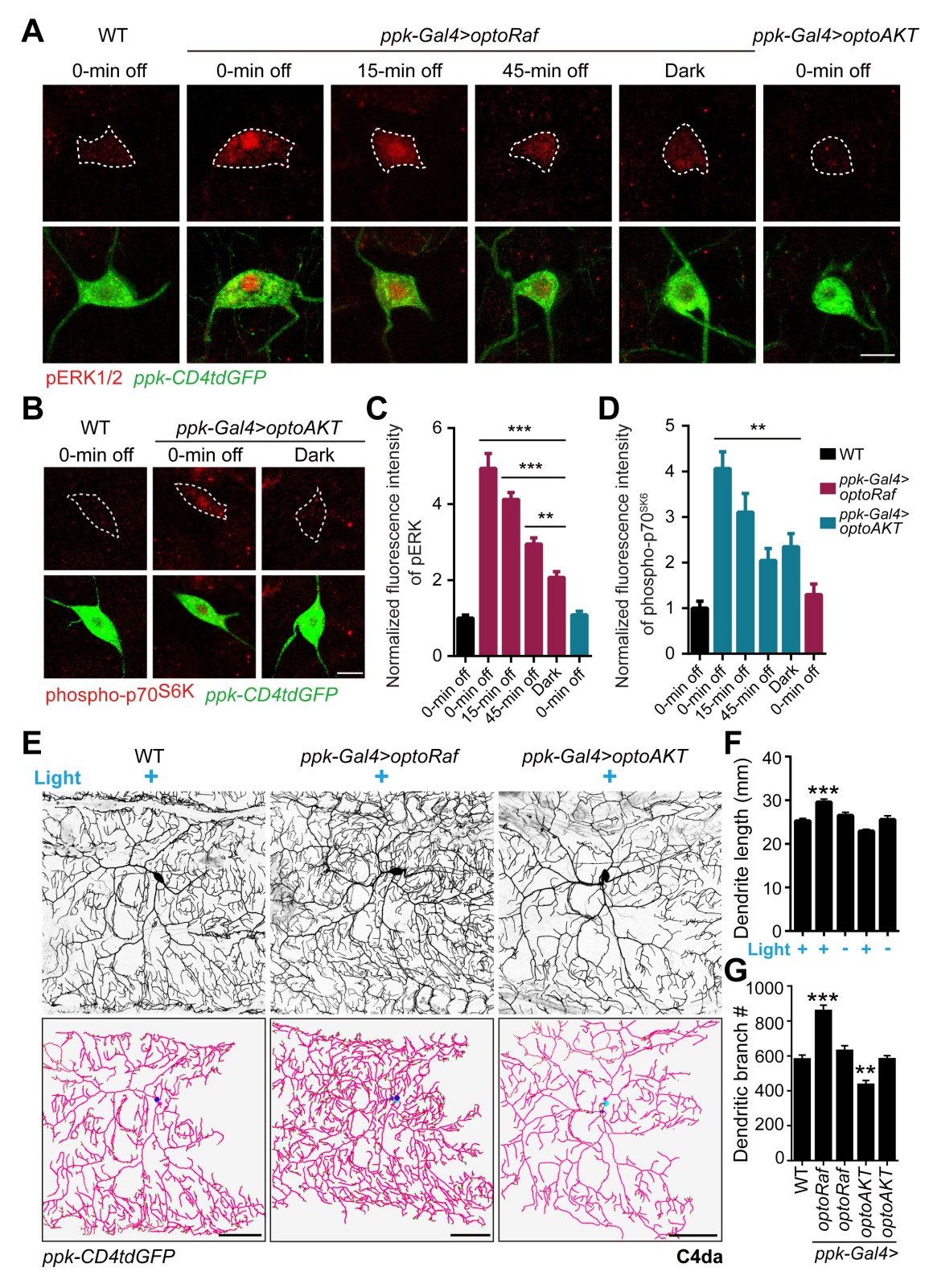

**Figure 2.** Activation of optoRaf but not optoAKT increases C4da neuron dendrite complexity. (A–D) 15 min blue light stimulation activates optoRaf and optoAKT in flies in vivo. After the light was off, the downstream effectors inactivated gradually. (A) The body walls from WT and optoRaf expressing larvae were dissected and stained for pERK1/2. The 15 min continuous light illumination leads to the enhanced fluorescent intensity and nuclear translocation of pERK in the optoRaf-expressing C4da neurons (labeled by *ppk-CD4tdGFP*). pERK signal is significantly increased even at 45 min after

*Figure 2 continued on next page*

*Figure 2 continued*

the light was off. Notably, the ERK signaling is not activated by light stimulation in optoAKT-expressing neurons. C4da neuron cell bodies are outlined by dashed white lines. Scale Bar = 10 µm. (**B**) Phospho-p70^S6K is activated by light illumination in optoAKT expressing neurons, and gradually returned to baseline after blue light was shut off. (**C**) Qualification of pERK fluorescence intensity in (**A**). The intensity of pERK in transgenic larvae was normalized to that of WT. WT (0 min off) *N* = 19, *optoRaf* (0 min off) *N* = 16, *optoRaf* (15 min off) *N* = 19, *optoRaf* (45 min off) *N* = 18, *optoRaf* (dark) *N* = 18, *optoAKT* (0 min off) *N* = 19 neurons. (**D**) Qualification of phospho-p70^S6K fluorescence intensity in (**B**). The intensity of phospho-p70^S6K in transgenic larvae was normalized to that of WT. WT (0 min off) *N* = 18, *optoAKT* (0 min off) *N* = 19, *optoAKT* (15 min off) *N* = 23, *optoAKT* (45 min off) *N* = 20, *optoAKT* (dark) *N* = 16, *optoRaf* (0 min off) *N* = 23 neurons. (**E–G**) Activation of Raf/MEK/ERK but not AKT signaling by 72 hr' light stimulation increases dendrite outgrowth and branching in C4da neurons. (**E**) Representative images of C4da neurons from WT, optoRaf and optoAKT expressing larvae with 72 hr' light stimulation and the unstimulated controls. Neurons were reconstructed with Neuronstudio. Scale bar = 50 µm. (**F**) Quantification of total dendrite length of C4da neurons. (**G**) Qualification of dendritic branch number. WT (light) *N* = 21, *optoRaf* (light) *N* = 21, *optoRaf* (dark) *N* = 21, *optoAKT* (light) *N* = 20, *optoAKT* (dark) *N* = 20 neurons. All data are mean ± SEM. The data were analyzed by one-way ANOVA followed by Dunnett's multiple comparisons test, \*\*p<0.01, \*\*\*p<0.001. See also *Figure 3—figure supplements 1–2*.

The online version of this article includes the following source data and figure supplement(s) for figure 2:

**Source data 1.** Activation of optoRaf but not optoAKT increases C4da neuron dendrite complexity.
**Figure supplement 1.** Activation kinetics of optoRaf and optoAKT in fly sensory neurons.
**Figure supplement 1—source data 1.** Activation kinetics of optoRaf and optoAKT in fly sensory neurons.
**Figure supplement 2.** The specific activation of ERK/p70^S6K in C4da neurons.
**Figure supplement 3.** Inactivation kinetics of optoRaf, and activation of optoRaf does not upregulate phospho-p70^S6K.

activity than the dark control. The difference in the inactivation kinetics may reflect distinct signaling sensitivity between Raf and AKT in optoRaf and optoAKT, respectively. These results confirmed that the intermittent pattern of light stimulation could modulate the temporal profile of ERK and AKT signaling activities.

We next investigated if optoRaf or optoAKT activation would affect neural development such as dendrite morphogenesis. We labeled C4da neurons with *ppk-CD4tdGFP* and reconstructed the dendrites of the lateral C4da neurons – v'ada. Without light stimulation, the dendrite complexity of neurons in transgenic larvae was comparable to that of WT (*Figure 2F and G*). However, optoRaf activation resulted in a significant increase in both total dendrite length and branch number, whereas optoAKT activation exhibited a slight reduction in dendritic branching (*Figure 2E–G*). These results confirm the possibility of independently activating the Raf/MEK/ERK and AKT signaling pathways in flies with our optogenetic tools, prompting us to test the feasibility of their in vivo applications, such as promoting axon regeneration with high spatial and temporal resolution.

## Activation of optoRaf or optoAKT results in enhanced axon regeneration in the PNS

Administration of neurotrophins to damaged peripheral neurons results in functional regeneration of sensory axons into the adult spinal cord in rat (*Ramer et al., 2000*). Here, our photoactivatable transgenic flies empower precise spatiotemporal control of the neurotrophic signaling in live animals. To test whether light-mediated activation of the Raf/MEK/ERK or AKT signaling subcircuits would also promote axon regrowth, we used a previously described *Drosophila* da sensory neuron injury model (*Song et al., 2012*; *Song et al., 2015*). Da neurons have been shown to possess distinct regeneration capabilities among different sub-cell types, and between the PNS and CNS, resembling mammalian neurons (*Song et al., 2012*; *Song et al., 2015*). In particular, the C4da neurons regenerate their axons robustly after peripheral injury, while the C3da neurons largely fail to regrow. Moreover, the axon regeneration potential of C4da neurons is also diminished after CNS injury. First, we asked whether optoRaf or optoAKT activation can enhance axon regeneration in the regeneration-competent C4da neurons in the PNS. We severed the axons of C4da neurons (labeled with *ppk-CD4tdGFP*) with a two-photon laser at 72 hr AEL, verified axon degeneration at 24 hr after injury (AI) and assessed axon regeneration at 48 hr AI. At this time point, about 79% C4da neurons in WT showed obvious axon regrowth, and the regeneration index (*Song et al., 2012*; *Song et al., 2015*), which refers to the increase in axon length normalized to larval growth (*Figure 3—figure supplement 1A and B*, and Materials and methods), was 0.3810 ± 0.06653 (*Figure 3A–C*). Strikingly, C4da neurons expressing optoRaf or optoAKT showed further enhanced regeneration potential in response to blue light, leading to a significant increase in the regeneration index (*optoRaf*: 0.7102 ± 0.1033;

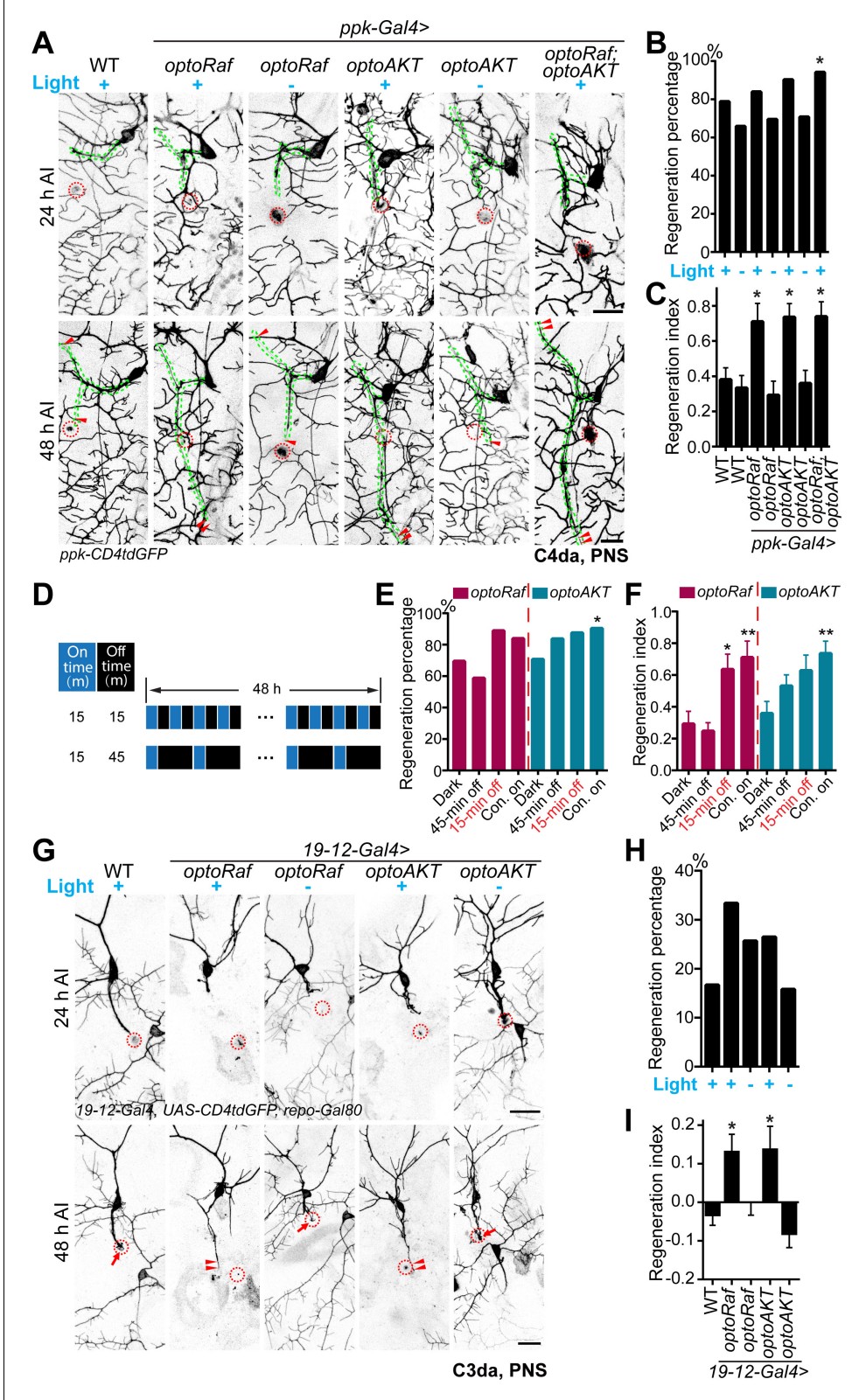

**Figure 3.** Light-stimulated optoRaf or optoAKT enhances axon regeneration in the PNS. (A–C) Compared with WT, C4da neurons expressing optoRaf or optoAKT show significantly increased axon regeneration in response to blue light. No enhancement was observed in the unstimulated controls. (A) C4da neuron axons were severed and their regeneration was assayed at 48 hr AI. The injury sites are marked by the red dashed circles and regenerating axons are marked by arrowheads. Axons are outlined with dashed green lines. Scale bar = 20 μm. (B) The regeneration percentage of

*Figure 3 continued on next page*

*Figure 3 continued*

light-stimulated transgenic groups is not significantly higher than WT. Fisher's exact test, p=0.3010, p=0.7608, p=0.4219, p=0.2007, p=0.5933, p=0.0440. (C) Qualification of C4da neuron axon regeneration by the regeneration index. WT (light) *N* = 33, WT (dark) *N* = 41, *optoRaf* (light) *N* = 36, *optoRaf* (dark) *N* = 36, *optoAKT* (light) *N* = 52, *optoAKT* (dark) *N* = 41, *optoRaf + optoAKT* (light) *N* = 51 neurons. Data are mean ± SEM, analyzed by one-way ANOVA followed by Dunnett's multiple comparisons test. (D–F) After injury, larvae were subjected to programmed light and dark cycles for a total of 48 hr. The intermittent light stimulation promotes axon regrowth in optoRaf expressing larvae similar to constant light when the off-time is 15 min. (D) The intermittent patterns of the light stimulus. (E) Compared with larvae incubated in dark, light stimulation is capable of increasing the percentage of regenerated axons. Fisher's exact test, p=0.3616, p=0.0795, p=0.2668, p=0.2030, p=0.1004, p=0.0285. (F) Qualification of C4da axon regeneration by the regeneration index. *OptoRaf* (dark) *N* = 36, *optoRaf* (45 min off) *N* = 46, *optoRaf* (15 min off) *N* = 36, *optoRaf* (Con. on) *N* = 36, *optoAKT* (dark) *N* = 41, *optoAKT* (45 min off) *N* = 49, *optoAKT* (15 min off) *N* = 40, *optoAKT* (Con. on) *N* = 52 neurons. Data are mean ± SEM, analyzed by one-way ANOVA followed by Dunnett's multiple comparisons test. (G–I) Blue light stimulation significantly enhances axon regeneration in the regeneration-incompetent C3da neurons. (G) C3da neuron axon degeneration was verified at 24 hr AI and axon regeneration was assessed at 48 hr AI. The injury sites are marked by the dashed circles, regenerated axons are demarcated by arrowheads, and arrows mark non-regenerated axons. Scale bar = 20 μm. (H) The regeneration percentage is not significantly different. Fisher's exact test, p=0.1146, p=0.4155, p=0.3979, p=1.000. (I) Qualification of axon regeneration by the regeneration index. WT (light) *N* = 42, *optoRaf* (light) *N* = 36, *optoRaf* (dark) *N* = 39, *optoAKT* (light) *N* = 34, *optoAKT* (dark) *N* = 38 neurons. Data are mean ± SEM, analyzed by one-way ANOVA followed by Dunnett's multiple comparisons test. *p<0.05, **p<0.01. See also *Figure 3— figure supplements 1* and *2*.

The online version of this article includes the following source data and figure supplement(s) for figure 3:

**Source data 1.** Light-stimulated optoRaf or optoAKT enhances axon regeneration in the PNS.
**Figure supplement 1.** Quantification of axon regeneration in the PNS.
**Figure supplement 2.** Co-activation of optoRaf and optoAKT does not further promote axon regeneration in C3da neurons.
**Figure supplement 2—source data 1.** Co-activation of optoRaf and optoAKT does not further promote axon regeneration in C3da neurons.

*optoAKT*: 0.7354 ± 0.07755), while there was no difference between WT and unstimulated transgenic flies (*Figure 3A–C*). In order to test the potential synergy between optoRaf and optoAKT, we co-expressed both transgenes in C4da neurons. While there was a slight increase in the regeneration percentage, activation of both ERK and AKT pathways in the same neuron did not further increase the regeneration index (0.7387 ± 0.08390) (*Figure 3A–C*).

The light stimulation paradigm used in the aforementioned regeneration experiments was constant blue light applied immediately after the injury. We reason that intermittent light stimulation may provide insights into the signaling kinetics in vivo and finetune axon regeneration dynamics. Therefore, instead of constant blue light illumination, we delivered two sets of programmed light patterns to injured larvae, 15 min on-15 min off or 15 min on-45 min off per cycle for 48 hr (*Figure 3D*). We found that, for optoRaf-expressing C4da neurons, when the off-time was 15 min, the intermittent light stimulation was sufficient to accelerate axon regrowth, with the regeneration index (0.6352 ± 0.09627) significantly increased compared with larvae incubated in the dark (*Figure 3E and F*). However, when the off-time was 45 min, the intermittent light failed to promote axon regeneration (*Figure 3E and F*). Considering that pERK activity remains slightly higher than the basal level after 45 min dark incubation (*Figure 2C*), the regeneration failure at 45 off-time suggests that optoRaf regulates C4da axon regeneration in a threshold-gated manner. On the other hand, C4da neurons expressing optoAKT displayed a graded response: a moderate increase of regeneration index (0.6278 ± 0.09801) in response to the 15 min on-15 min off light and a smaller uptick (0.5312 ± 0.06963) to the 15 min on-45 min off light; both were less effective than the constant light stimulation (*Figure 3E and F*). These results suggest that although the higher frequency of light stimulation generally resulted in stronger regeneration potential in the transgenic flies, constant light was not always required for maximum axon regeneration. Moreover, optoRaf and optoAKT differ in their signaling kinetics during regeneration, showing a gated versus graded response, respectively.

We next determined whether optoRaf or optoAKT activation would trigger regeneration in C3da neurons, which are normally incapable of regrowth (*Song et al., 2012*). C3da neurons were labeled with *19–12-Gal4, UAS-CD4tdGFP, repo-Gal80* and injured using the same paradigm as C4da neurons. Compared with WT, which exhibited poor axon regeneration ability demonstrated by the low regeneration percentage and the negative regeneration index (−0.03201 ± 0.02752) (*Figure 3G–I*), light stimulation significantly increased the regeneration index in optoRaf- or optoAKT-expressing larvae to 0.1298 ± 0.04637 or 0.1354 ± 0.06161, respectively (*Figure 3G–I*). Similar to C4da neurons, activation of both Raf/MEK/ERK and AKT pathways failed to further enhance axon regrowth compared to optoRaf or optoAKT activation alone (*Figure 3—figure supplement 2*). This result confirms

that the actions of optoRaf and optoAKT are not additive in promoting axon regeneration, suggesting that these two subcircuits may share the same downstream components in neuroregeneration (see Discussion). Altogether, these data indicate that optoRaf and optoAKT activation not only accelerates axon regeneration but also converts regeneration cell-type specificity.

## Spatial activation of optoRaf or optoAKT improves pathfinding of regenerating axons

While C4da neurons are known to possess the regenerative potential, it is unclear whether the regenerating axons navigate correctly. To address this question, we focused on v'ada – the lateral C4da neurons. Uninjured v'ada axons grow ventrally, showing a typical turn and then join the axon bundle with the ventral C4da neurons (*Figure 3—figure supplement 1A*). We found that their regenerating axons preferentially regrew away from the original ventral trajectory (*Figure 4A and B* white bars). More than 60% v'ada axons bifurcated and formed two branches targeting opposite directions. In the majority cases in WT, the ventral branch, which extends toward the correct trajectory, regenerated less frequently than the dorsal branch, with 15% v'ada containing only the ventral branch (*Figure 4A and B* black bars). One possibility is that the ventral branch encounters the injury site, which may retard its elongation. As a result, only a minority of regenerating axons are capable of finding the correct path. The poor pathfinding of regenerating axons was similar among WT and the transgenic larvae, regardless of whether incubated with whole-field light or in the dark (*Figure 4B*). Thus, proper guidance of the regenerating axons toward the correct trajectory remained to be resolved.

We thus investigated whether spatially restricted activation of the neurotrophic signaling using our optogenetic system could guide the regenerating axons. To specifically enhance the regrowth of the ventral branch, we used a confocal microscope to focus the blue light (delivered by the 488 nm argon-ion laser) on the ventral branch for 5 min at 24 hr AI. The lengths of both the ventral and dorsal branches were measured at 24 hr AI and 48 hr AI. We subtracted the increased dorsal branch length ($\Delta$dorsal) from the increased ventral branch length ($\Delta$ventral), then divided that by the total increased length of these two branches (*Figure 4D*). This value was defined as the relative regeneration ratio. If the dorsal branch exhibits more regenerative potential, the ratio would be negative; otherwise, it would be positive. Without light stimulation, the relative regeneration ratio of the transgenic larvae (*optoRaf*: $-0.6062 \pm 0.1453$; *optoAKT*: $-0.5530 \pm 0.1011$) was comparable to that of WT ($-0.5786 \pm 0.08229$) (*Figure 4C and D*), confirming preferred regrowth of the dorsal branch. Strikingly, the 5 min local blue light stimulation significantly increased the ratio in optoRaf- or optoAKT-expressing v'ada (*optoRaf*: $0.04762 \pm 0.1123$; *optoAK*T: $-0.1725 \pm 0.09560$), while this transient stimulation resulted in no difference in WT ($-0.6018 \pm 0.1290$) (*Figure 4C and D*). This result indicates that a single pulse of local light stimulation was sufficient to lead to preferential regrowth of the ventral branch. Notably, although whole-field light illumination could significantly promote axon regrowth, it failed to increase the relative regeneration ratio in transgenic larvae (Con. on *optoRaf*: $-0.7048 \pm 0.1015$; Con. on *optoAKT*: $-0.5517 \pm 0.09644$) (*Figure 4D*), revealing the difference between activating the neurotrophic signaling in a whole neuron and a single lesioned axon branch. On the other hand, while a 5-min local light stimulation did not lead to an overall enhancement of axon regrowth, it provided adequate guidance instructions for the regenerating axons to make the correct choice.

## Activation of optoRaf or optoAKT promotes axon regeneration and functional recovery in the CNS

Achieving functional axon regeneration after CNS injury remains a major challenge in neural repair research. Motivated by the capacity of optoRaf and optoAKT to accelerate axon regeneration in the PNS, we went on to determine whether they also show efficacy after CNS injury. We focused on the axons of C4da neurons, which project into the ventral nerve cord (VNC) and form a ladder-like structure. Each pair of axon bundles correspond to one body segment in an anterior-posterior pattern (*Li et al., 2020*). We injured the abdominal A6 and A3 bundles by laser as previously described (*Song et al., 2012*; *Li et al., 2020*; *Figure 5—figure supplement 1*), and confirmed axon degeneration at 24 hr AI (*Figure 5A*). At 48 hr AI, we found that axons began to extend from the retracted axon stem and towards the commissure region. We defined a commissure segment as regenerated

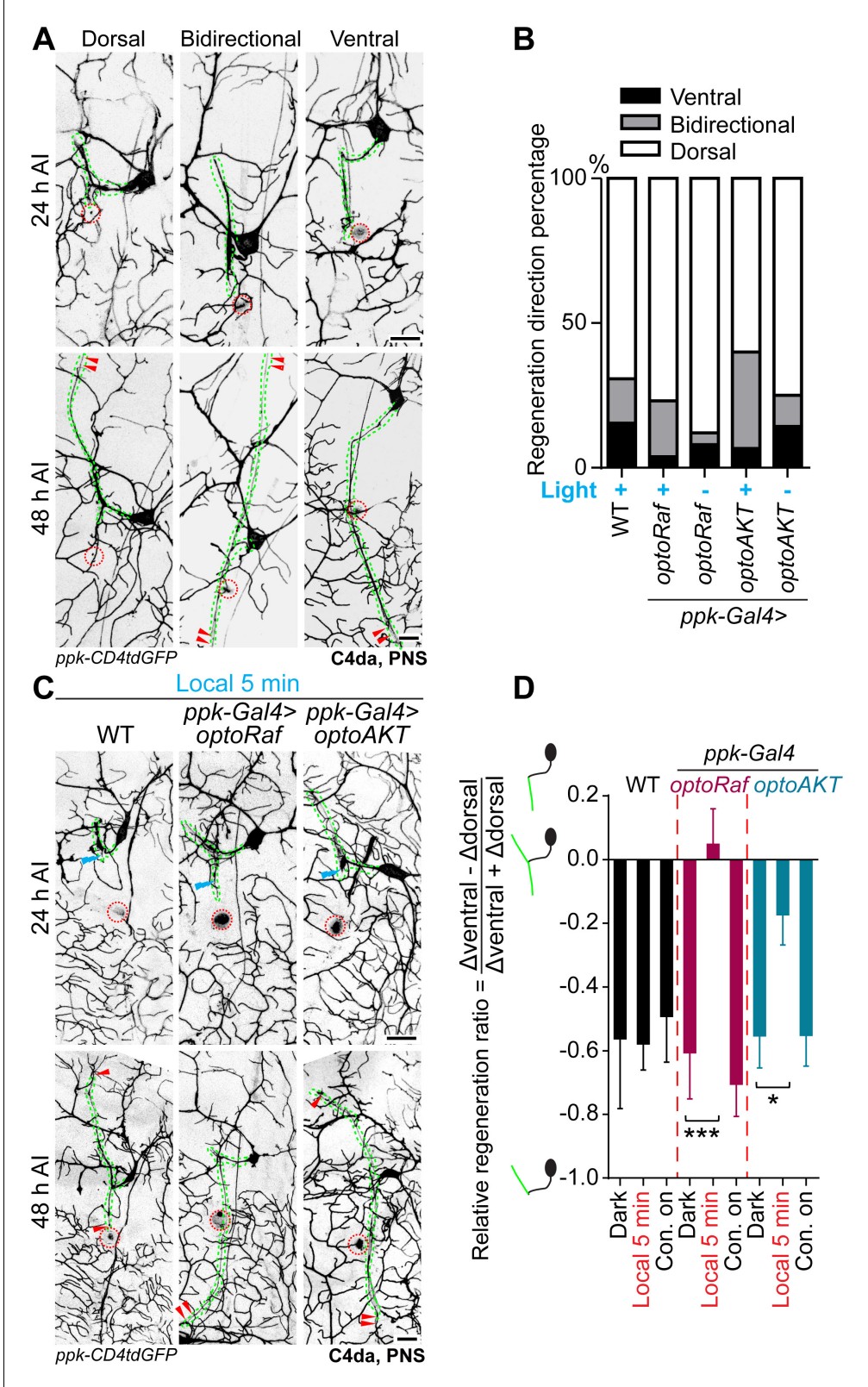

**Figure 4.** Local optogenetic stimulation conveys guidance instructions to regenerating axons. (A, B) Regenerating axons prefer to regrow away from the original trajectory, with only a minority of axons finding the correct path. (A) Representative images of axons retracting or bifurcating at 24 hr AI. At 48 hr AI in WT, regenerating axons extend dorsally, ventrally, or both directions. The injury site is marked by the dashed circles and regenerating axons are marked by arrowheads. Axons are outlined with dashed green lines. Scale bar = 20 μm. (B) Light stimulation fails to increase the percentage of

*Figure 4 continued on next page*

*Figure 4 continued*

axons regrowing towards the right direction. The percentage of axons extending towards the correct trajectory (ventral + bidirectional) were analyzed by Fisher's exact test, p=0.7554, p=0.1729, p=0.6097, p=0.7638. WT (light) $N$ = 26, *optoRaf* (light) $N$ = 26, *optoRaf* (dark) $N$ = 25, *optoAKT* (light) $N$ = 45, *optoAKT* (dark) $N$ = 28 neurons. (C, D) Restricted local activation of optoRaf or optoAKT significantly increases the relative regeneration ratio. The ratio is defined to weigh the regeneration potential of the ventral branch against the dorsal branch. (C) A single pulse of light stimulation delivered specifically on the ventral axon branch at 24 hr AI (blue flash symbol) is capable of promoting the preferential extension of regenerating axons in optoRaf or optoAKT expressing larvae. The injury sites are demarcated by the dashed red circles and regenerating axons are marked by arrowheads. Axons are outlined with dashed green lines. Blue flash symbols show the restrict light delivery to the ventral branch. (D) Qualification of the relative regeneration ratio of v'ada. WT (dark) $N$ = 32, WT (local 5 min) $N$ = 32, WT (Con. on) $N$ = 33, *optoRaf* (dark) $N$ = 32, *optoRaf* (local 5 min) $N$ = 35, *optoRaf* (Con. on) $N$ = 36, *optoAKT* (dark) $N$ = 33, *optoAKT* (local 5 min) $N$ = 34, *optoAKT* (Con. on) $N$ = 36 neurons. Data are mean ± SEM, analyzed by one-way ANOVA followed by Dunnett's multiple comparisons test, *p<0.05, ***p<0.001.

The online version of this article includes the following source data for figure 4:

**Source data 1.** Local optogenetic stimulation conveys guidance instructions to regenerating axons.

only when at least one axon extended beyond the midline of the commissure region or joined into other intact bundles (*Figure 5—figure supplement 1*). In WT, only 16% of lesioned commissure segments displayed obvious signs of regrowth (*Figure 5A and B*). To quantify the extent of regrowth, we measured the length of the regrown axons and normalized that to the length of a commissure segment – regeneration index (*Figure 5—figure supplement 1*, Materials and methods). After light stimulation, the regeneration indexes of the two transgenic lines (*optoRaf*: 5.375 ± 0.3391; *optoAKT*: 4.765 ± 0.4236) were significantly increased compared with the WT control (2.643 ± 0.3050), and the percentage of regenerating commissure segments also exhibited a mild increase in both optoRaf- and optoAKT- expressing larvae (*Figure 5A–C*). On the other hand, there was no significant difference between WT and the unstimulated transgenic flies (*Figure 5A–C*). This result suggests that both signaling subcircuits reinforce C4da neuron axon regeneration in the CNS.

We then tested whether the axon regrowth in the CNS induced by optoRaf or optoAKT activation leads to behavioral improvement. We utilized a recently established behavioral recovery paradigm based on larval thermonociception (*Figure 6A*, Materials and methods) (*Li et al., 2020*). In brief, we injured the A7 and A8 C4da neuron axon bundles in the VNC, which correspond to the A7 and A8 body segments in the periphery. We then assessed the nociceptive behavior in these larvae in response to a 47°C heat probe applied at the A7 or A8 segments at 24 and 48 hr AI. Since C4da neurons are essential for thermonociception, injuring A7 and A8 axon bundles in the VNC would lead to an impaired nociceptive response to the heat probe specifically at body segments A7 and A8. Indeed, all the injured larvae exhibited diminished response at 24 hr AI, while the total score is approaching three in uninjured WT larvae (*Figure 6B*). At 48 hr AI, substantial recovery was observed in the two transgenic groups with light stimulation, whereas WT showed a very limited response and a low recovery percentage (*Figure 6B and C*). Both the response score and the percentage of larvae exhibiting behavioral recovery in these two groups were more than twice as that of the WT, while the unstimulated groups were comparable to WT. Altogether, these results demonstrate that our optogenetic system empowers ligand-free and non-invasive control of the Raf/MEK/ERK and AKT pathways in flies, which not only promote axon regeneration after injury but also benefit functional recovery, suggesting that the regenerated axons may rewire and form functional synapses.

## Discussion

Neurotrophins are known to activate Trk receptors and trigger the Raf/MEK/ERK, AKT, and PLCγ pathways which are involved in cell survival, neural differentiation, axon and dendrite growth and sensation (*Bibel and Barde, 2000*; *Huang and Reichardt, 2001*; *Chao, 2003*; *Cheng et al., 2011*; *Joo et al., 2014*). Here, we used optogenetic systems to achieve specific and reversible activation of the neurotrophin subcircuits, including the Raf/MEK/ERK (via optoRaf) and AKT (via optoAKT) signaling pathways. We further verified that optoRaf and optoAKT did not show crosstalk at the level of phosphorylated ERK and AKT proteins, and activation of optoRaf but not optoAKT promoted PC12 cell differentiation.

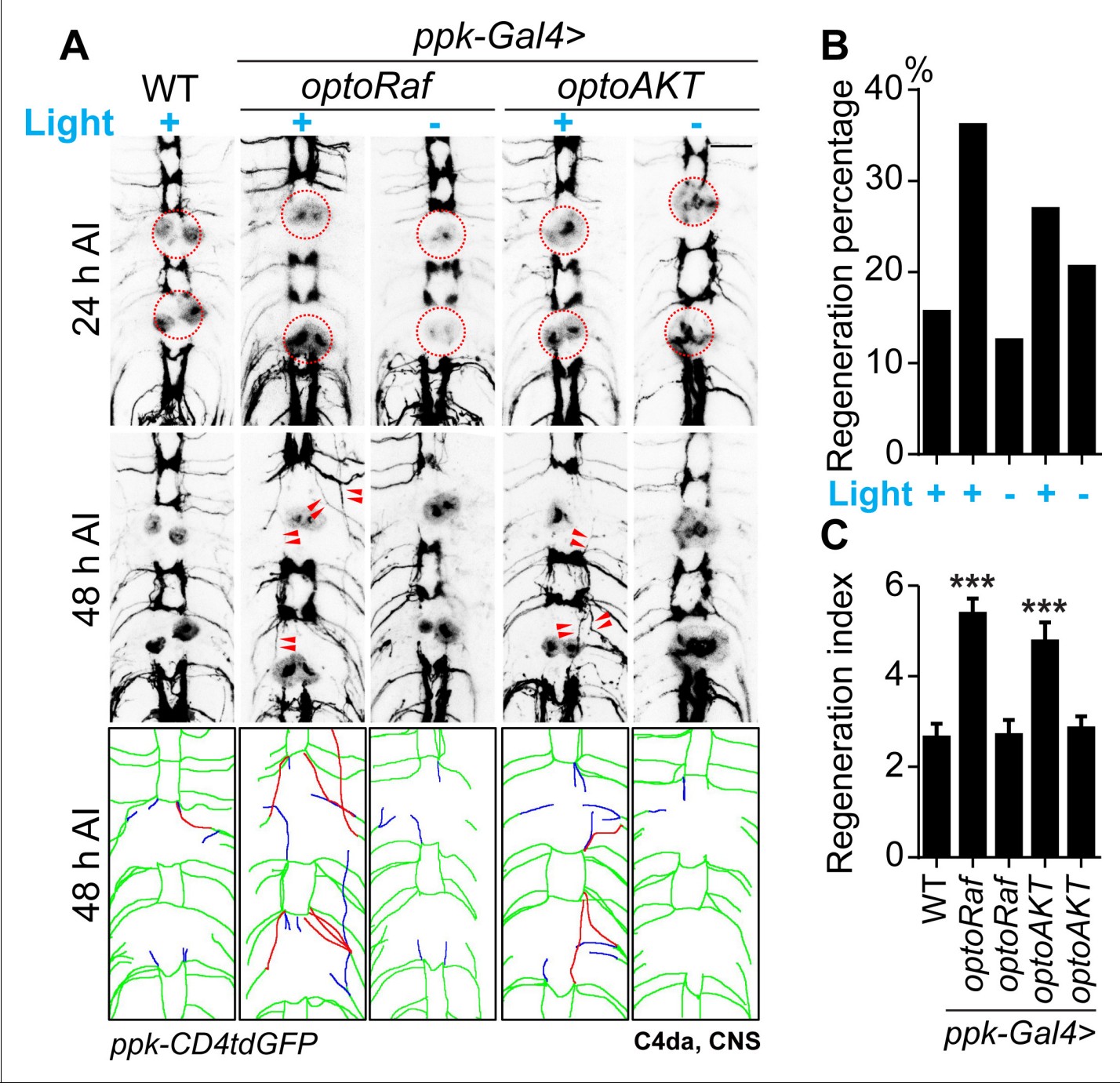

**Figure 5.** Activation of optoRaf or optoAKT promotes axon regeneration in the CNS. (**A–C**) Light stimulation significantly enhances axon regeneration in the VNC of optoRaf- or optoAKT-expressing larvae. (**A**) Complete degeneration in A3 and A6 commissure segments was confirmed at 24 hr AI and regeneration of these two segments was assayed independently at 48 hr AI. The injury sites are marked by the red dashed circles and regenerating axons are labeled by arrowheads. In the schematic diagrams, regrowing axons that reached other bundles and thus define a regenerating commissure segment are highlighted in red, while other regrowing axons are illustrated in blue. Scale bar = 20 µm. (**B**) The regeneration percentage is not significantly different. Fisher's exact test, p=0.0971, p=1.000, p=0.3415, p=0.7524. (**C**) Qualification of axon regeneration in VNC by the regeneration index. WT (light) *N* = 32, *optoRaf* (light) *N* = 36, *optoRaf* (dark) *N* = 32, *optoAKT* (light) *N* = 26, *optoAKT* (dark) *N* = 34 segments. Data are mean ± SEM, analyzed by one-way ANOVA followed by Dunnett's multiple comparisons test, ***p<0.001. See also *Figure 5—figure supplement 1*.

The online version of this article includes the following source data and figure supplement(s) for figure 5:

**Source data 1.** Activation of optoRaf or optoAKT promotes axon regeneration in the CNS.
**Figure supplement 1.** Quantification of axon regeneration in the CNS.

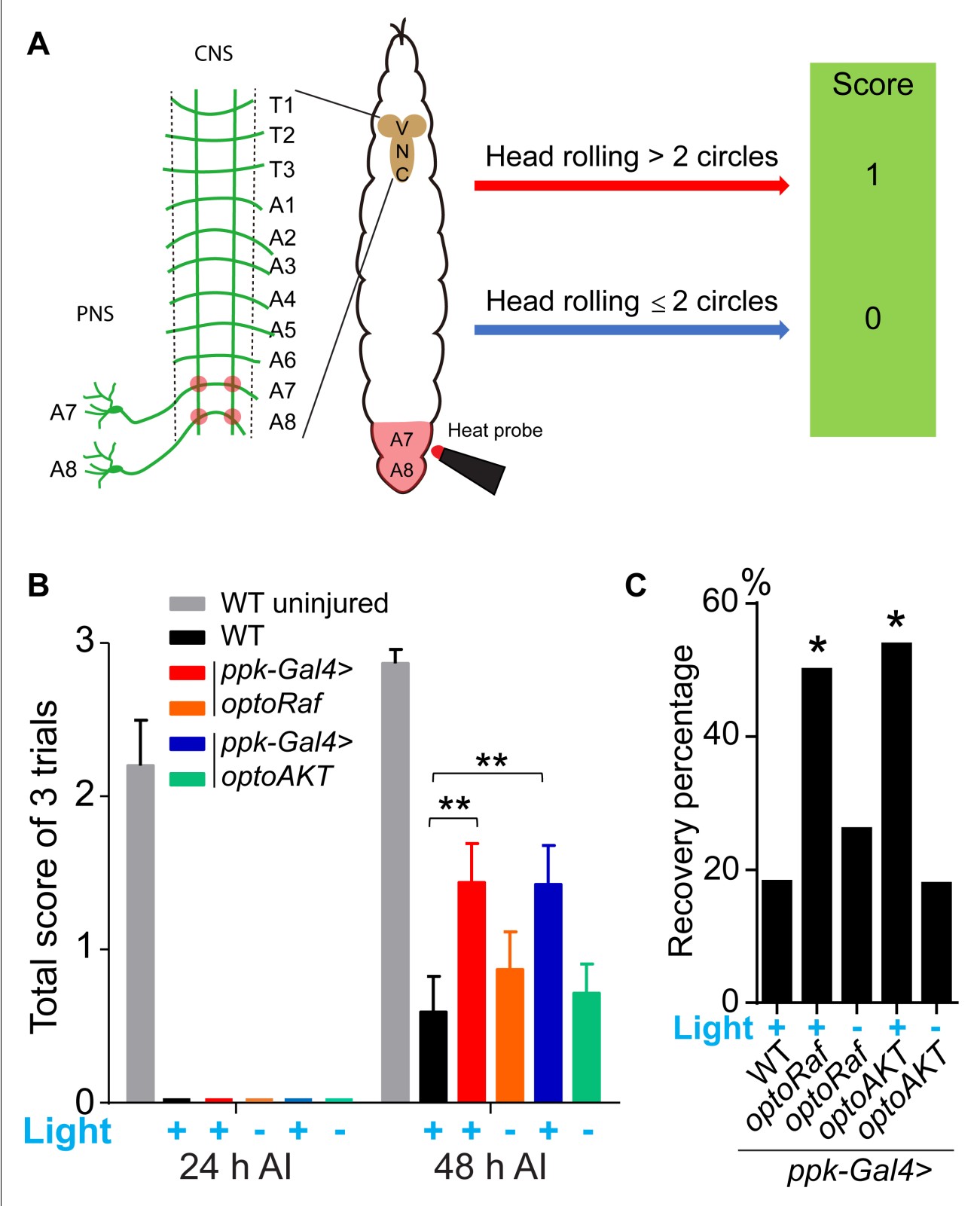

**Figure 6.** Activation of optoRaf or optoAKT promotes functional regeneration after CNS injury. (**A**) A schematic drawing of the behavioral recovery test. The A7 and A8 C4da neuron axon bundles (corresponding to the A7 and A8 body segments) in the VNC were injured by laser and the larva was then subjected to three consecutive trials at 24 and 48 hr AI, respectively. In each trial, a 47°C heat probe was applied at the A7 or A8 segments. A fully recovered larva would produce a stereotypical rolling behavior in response to the heat probe and would be scored as '1', otherwise as '0'. If the total

*Figure 6 continued on next page*

Figure 6 continued

score of the three trials was below 1 at 24 hr AI but increased to 2 or 3 at 48 hr AI, the larva was defined as recovered. (B, C) The behavioral recovery test was performed at 24 hr and 48 hr after VNC injury (A7 and A8 bundles). Larvae expressing optoRaf or optoAKT exhibit significantly accelerated recovery in response to light stimulation. (B) Qualification of the total scores at each time point. WT (uninjured) N = 15, WT (light) N = 22, optoRaf (light) N = 32, optoRaf (dark) N = 23, optoAKT (light) N = 26, optoAKT (dark) N = 28. Data are mean ± SEM, analyzed by two-way ANOVA followed by Tukey's multiple comparisons test. (C) Qualification of the recovery percentage. The data were analyzed by Fisher's exact test, p=0.0230, p=0.7222, p=0.0167, p=1.000. *p<0.05, **p<0.01.

The online version of this article includes the following source data for figure 6:

**Source data 1.** Activation of optoRaf or optoAKT promotes functional regeneration after CNS injury.

Note that in the canonical growth factor signaling pathways, crosstalk actually occurs between the ERK and AKT signaling pathways, particularly at the upstream signaling node such as Ras. Indeed, the binding of growth factors to their receptors activates the transmembrane receptor tyrosine kinase, which recruits adaptor protein such as Grb2 (growth factor receptor-bound protein) and Sos (son of sevenless), a guanine exchange factor (GEF) for Ras. Sos then transforms the inactive, GDP-bound Ras to an active, GTP-bound Ras, which then recruits multiple proteins, including Raf and PI3K, an upstream kinase for AKT, to the plasma membrane. Thus, Ras serves as a common signaling node and therefore creates possible signaling crosstalk between the PI3K-AKT and Raf-MEK-ERK pathway. Another possible signaling crosstalk arises from the PI3K-mediated production of phospholipids, which could recruit a number of signaling molecules containing the lipid-binding domain (e.g., PH domain) including Sos, which then affects Ras/Raf activation. Our observation that optoRaf and optoAKT do not crosstalk (i.e. optoRaf does not activate the AKT downstream effector p70$^{S6K}$; optoAKT does not activate pERK) may arise from the fact that both optoRaf and optoAKT bypass the ligand-binding, receptor activation, Ras activation, and phospholipid production signaling steps. Activation of optoRaf and optoAKT does require upstream signaling molecules (e.g. kinases).

However, there could be common downstream signaling molecules (such as transcription factors) that mediate the effects of neural regeneration by optoRaf and optoAKT. While ongoing efforts aim to elucidate these common signaling effectors, evidence from previous literature (some from other cell types) implies several possible candidates such as CREB (cAMP response element-binding protein) and FOXO (forkhead box transcription factors). Activation of Raf leads to phosphorylation of CREB, a family of transcription factors that regulate cell survival (*Ginty et al., 1994*). Evidence also suggests that CREB is a regulatory target for AKT (*Du and Montminy, 1998*). Besides the positive regulation of CREB by ERK and AKT signaling, their activity could also negatively regulate the function of FOXO transcription factors. FOXO is a family of transcription factors that can directly be phosphorylated by AKT (*Brunet et al., 1999*). Phosphorylated FOXO transcription factors translocate out of the nucleus, and their transcriptional program is attenuated. Interestingly, phosphorylated ERK can downregulate FOXO activity by directly interacting with and phosphorylates FOXO3a at Ser 294, Ser 344, and Ser 425, which leads to FOXO3a degradation via an MDM2-mediated ubiquitin-proteasome pathway (*Yang et al., 2008*). Additional evidence supporting this idea is that inhibition of PI3K/AKT and MEK/ERK pathways both enhance the activation of FOXO transcription factors in pancreatic cancer cells (*Roy et al., 2010*).

After spinal cord injury, the synthesis of neurotrophins is elevated to support axon regrowth (*Cho et al., 1998*; *Hayashi et al., 2000*; *Fukuoka et al., 2001*; *Fang et al., 2017*). AKT signaling, which functions downstream of Trk receptors, was reported to accelerate axon regeneration in mammals (*Guo et al., 2016*; *Miao et al., 2016*). While NGF family members of neurotrophic factors have only been identified in vertebrates, the AKT pathway has also been shown to promote axon regrowth in flies (*Song et al., 2012*). However, the role of Raf/MEK/ERK signaling during nerve repair is controversial. Although some studies revealed that ERK is involved in axon extension, others suggested that ERK activation impedes axon regeneration and functional recovery (*Markus et al., 2002*; *Huang et al., 2017*; *Cervellini et al., 2018*). To specifically evaluate the efficacy of Raf/MEK/ERK and AKT signaling in promoting axon regeneration, we generated fly strains with tissue-specific expression of optoRaf or optoAKT and found that light stimulation was sufficient to activate the corresponding downstream components in fly larvae in vivo. Consistent with previous studies (*He and Jin, 2016*), we found that AKT activation resulted in significantly increased axon regeneration in

C4da neurons as well as the regeneration-incompetent C3da neurons. Interestingly, we found that C4da and C3da neurons expressing optoRaf also exhibited greater regeneration potential in response to light stimulation. This result also corroborates with a previous finding that activated B-RAF signaling enables axon regeneration in the mammalian CNS (*O'Donovan et al., 2014*). We speculate that the differential outcomes of ERK activation on axon regeneration may be due to the different injury models used, and the strength and cell type origin of ERK signaling.

The regenerative capacity varies significantly among different neuronal subtypes, as well as the PNS and CNS. Although the administration of neurotrophins enhances axon regeneration in peripheral neurons, its capacity to promote functional regeneration in the CNS is limited, in part due to the inaccessibility of neurotrophins to reach injured axons (physical barrier) (*Silver and Miller, 2004*; *Yiu and He, 2006*) and innate inactivation of the regenerating program in CNS (*Lu et al., 2014*). OptoRaf and optoAKT could be used to address both issues by direct delivery of light (rather than ligand) to reactivate the regenerating program and thereby significantly increase neural regeneration in the CNS as well. We further showed that activation of the Raf/MEK/ERK or AKT subcircuit was capable of improving behavioral performance in fly larvae, suggesting that it may promote synapse regeneration leading to functional recovery.

Ineffective functional recovery at least partially results from the inappropriate pathfinding of the regenerative neurons. As shown in this study, the majority of regenerating C4da neuron axons preferentially grew away from their original trajectory. We surprisingly found that delivering a 5 min light stimulation to the ventral branch, which extended toward the correct direction, was sufficient to convey guidance instructions and increase the preferential elongation of the ventral branch against the dorsal branch. Correct guidance cannot be achieved by whole-body administration of pharmacological reagents. Similarly, when casting blue light on the whole transgenic larvae, light stimulation must be given at a high frequency to promote axon regrowth (there is a threshold for the light off-time), and the dorsal branch extension was also dominant in this case. This result highlights the importance and necessity of restricted activation of neurotrophic signaling. Indeed, the strength and location of Raf/MEK/ERK and AKT activation during axon regeneration may be important to the functional consequences. Notably, although the transient restricted stimulation likely affects the decision-making of the growth cone at the branching point, constant light is still required to increase overall axon regeneration.

Neurotrophins are engaged in a variety of important cellular processes, and their physiological concentration is essential for the normal function of both neurons and non-neuronal cells (*Rose et al., 2003*; *Xiao et al., 2010*; *Pöyhönen et al., 2019*). Despite exhibiting substantial efficacy for enhancing nerve regeneration, neurotrophin-based therapeutic applications have been confronted with a number of obstacles such as their nociceptive side effects and lack of strategy for localized signaling activation (*Aloe et al., 2012*; *Mitre et al., 2017*; *Mahar and Cavalli, 2018*; *Sung et al., 2019*). OptoRaf and optoAKT aim to improve neurotrophin signaling outcomes by preferentially activating the neuroregenerative program and enabling spatiotemporal control. Our systems offer insights into the ERK and AKT subcircuits and delineate their differential roles downstream of neurotrophin activation, as evidenced by the distinct functional outcomes of Raf/MEK/ERK and AKT signaling in several aspects. First, ERK signaling promoted PC12 cell neuritogenesis, which was not induced by AKT activation. Second, elevated ERK activity significantly increased dendritic complexity, while on the contrary, AKT activation led to decreased dendrite branching. Third, optoRaf and optoAKT displayed different sensitivity in response to light illumination when expressed in *Drosophila* C4da neurons. Correspondingly, neurons expressing optoRaf and optoAKT responded differently to intermitted light stimulation after injury, suggesting that the strength and activation duration of optoRaf and optoAKT is differentially gauged during axon regeneration. These collectively suggest that, since Raf could be activated by membrane translocation as well as dimerization, CRY2 oligomerization could further lead to a more potent Raf. This multimodal activation mechanism may render that a threshold of optoRaf can be reached so that a saturated ERK activation could be achieved. On the other hand, AKT activation does not depend on dimerization and may display a graded response. As a result, optoAKT activates the AKT pathway in a dose-dependent manner and may not recapitulate the maximum activation of AKT. This work provides a proof-of-concept to use optogenetics to accelerate and navigate axon regeneration in mammalian injury models. Besides spatiotemporal control of the neurotrophic signaling, optoRaf and optoAKT allow for finetuning of the signaling activity with programmed light pattern during axon regeneration.

Follow-up studies are warranted to determine how Raf/MEK/ERK and AKT subcircuits are involved in each process of nerve repair, including lesioned axon degeneration, regenerating axon initiation and extension, and the formation of new synapses and remyelination in mammals. Understanding the machinery will, in turn, allow better utilization and development of the optogenetic systems. Recently, Harris et al. succeeded in directing axon outgrowth with optogenetic tools in zebrafish embryos (*Harris et al., 2020*). Although intact optogenetics in larger mammals is limited by the poor penetration depth of blue light (less than 1 mm), we are excited to witness the rapid progress in implantable, wireless µLED devices (*Jeong et al., 2015*; *Park et al., 2015b*) and the integration of optogenetics with long-wavelength-responsive nanomaterials such as the upconversion nanoparticles (*He et al., 2015*; *Wu et al., 2016*; *Chen et al., 2018*), both of which would facilitate precise delivery of light stimulation.

# Materials and methods

## Key resources table

| Reagent type (species) or resource | Designation | Source or reference | Identifiers | Additional information |
|---|---|---|---|---|
| Gene (*Homo sapiens*) | *Raf1* | PMID:24667437 | NCBI Gene ID: 207 | |
| Gene (*Homo sapiens*) | *AKT* | PMID:27082641 | NCBI Gene ID: 5894 | |
| Strain, strain background (*D. melanogaster*) | *19–12-Gal4* | PMID:21068723 | FLYB: FBti0148308 | FlyBase symbol: P{GAL4}19–12 |
| Strain, strain background (*D. melanogaster*) | *repo-Gal80* | PMID:19091965 | FLYB: FBtp0067904 | Flybase symbol: P{repo-GAL80.L} |
| Strain, strain background (*D. melanogaster*) | *ppk-CD4-tdGFP* | PMID:21606367 | BDSC: 35842 FLYB: FBti0143429 RRID:BDSC_35842 | Flybase symbol: P{ppk-CD4-tdGFP}1b |
| Strain, strain background (*D. melanogaster*) | *ppk-Gal4* | PMID:21606367 | BDSC: 32079 FLYB: FBti0131208 RRID:BDSC_32079 | Flybase symbol: P{ppk-GAL4.G}3 |
| Strain, strain background (*D. melanogaster*) | *UAS-optoRaf* | This paper | | Transgenic fly with inducible expression of optoRaf, Dr. Yuanquan Song |
| Strain, strain background (*D. melanogaster*) | *UAS-optoAKT* | This paper | | Transgenic fly with inducible expression of optoAKT, Dr. Yuanquan Song |
| Cell line (*Homo sapiens*) | HEK293T | PMID:32277988 | | Dr. Lin-Feng Chen |
| Cell line (*Rattus norvegicus*) | PC12 | PMID:22206868 | | Dr. Tobias Meyer |
| Cell line (*Mesocricetus auratus*) | BHK-21 | PMID:26080442 | | Dr. Steven Chu |
| Transfected construct (*Homo sapiens*) | pEGFP N1 CRY2-mCh-Raf1-P2A-CIBNx2-EGFP-CAAX | This paper | | OptoRaf, CIBN-CRY2 based Raf membrane translocation system. Dr. Kai Zhang |

*Continued on next page*

*Continued*

| Reagent type (species) or resource | Designation | Source or reference | Identifiers | Additional information |
|---|---|---|---|---|
| Transfected construct (*Homo sapiens*) | pEGFP N1 CRY2-mCh-Raf1 S338A-P2A-CIBNx2-EGFP-CAAX | This paper | | OptoRaf S338A CIBN-CRY2 based Raf S338A membrane translocation system. Dr. Kai Zhang |
| Transfected construct (*Homo sapiens*) | pEGFP N1 CRY2-mCh-AKT-P2A-CIBNx2-EGFP-CAAX | This paper | | OptoAKT, CIBN-CRY2 based AKT membrane translocation system. Dr. Kai Zhang |
| Transfected construct (*Arabidopsis thaliana*) | pEGFP C1 CIBN-EGFP-CAAX | PMID:21037589 | | Dr. Chandra Tucker |
| Transfected construct (*Homo sapiens*) | pEGFP C1 Raf1-EGFP-CaaX | PMID:24667437 | | A constitutively active form of Raf1. Dr. Kai Zhang |
| Transfected construct (*Rattus norvegicus*) | pEGFP N1 ERK2-EGFP | PMID:24667437 | | Fluorescently label ERK2 for nuclear translocation assay. Dr. Kai Zhang |
| Transfected construct (*Homo sapiens*) | pEGFP N1 FOXO3-EGFP | PMID:10102273 | | Fluorescently labeled FOXO3 for nuclear export assay. Dr. Kai Zhang |
| Antibody | Anti-Phospho-p44/42 MAPK (Erk1/2) (Thr202/Tyr204), rabbit monoclonal | Cell Signaling Technology | Cat# 4370T RRID:AB_2315112 | IF (1:400) |
| Antibody | Anti-Phospho-*Drosophila* p70 S6 Kinase (Thr398), rabbit polyclonal | Cell Signaling Technology | Cat# 9209S RRID:AB_2269804 | IF (1:100) |
| Antibody | Phospho-p44/42 MAPK (Erk1/2) (Thr202/Tyr204), rabbit polyclonal | Cell Signaling Technology | Cat# 9101S RRID:AB_331646 | WB (1:1000) |
| Antibody | p44/42 MAPK (Erk1/2), rabbit polyclonal | Cell Signaling Technology | Cat# 9102S RRID:AB_330744 | WB (1:1000) |
| Antibody | c-Raf, rabbit polyclonal | Cell Signaling Technology | Cat# 9422S RRID:AB_390808 | WB (1:1000) |
| Antibody | Phospho-c-Raf (Ser338) (56A6), rabbit monoclonal | Cell Signaling Technology | Cat# 9427S RRID:AB_2067317 | WB (1:1000) |
| Antibody | Phospho-Akt (Ser473) (D9E) XP, rabbit monoclonal | Cell Signaling Technology | Cat# 4060S RRID:AB_2315049 | WB (1:1000) |

*Continued on next page*

*Continued*

| Reagent type (species) or resource | Designation | Source or reference | Identifiers | Additional information |
|---|---|---|---|---|
| Antibody | Phospho-Akt (Thr308) (244F9), rabbit monoclonal | Cell Signaling Technology | Cat# 4056S RRID:AB_331163 | WB (1:1000) |
| Antibody | Akt (pan) (C67E7), rabbit monoclonal | Cell Signaling Technology | Cat# 4691S RRID:AB_915783 | WB (1:1000) |
| Antibody | PLCγ1, rabbit polyclonal | Cell Signaling Technology | Cat# 2822S RRID:AB_2163702 | WB (1:1000) |
| Antibody | Phospho-PLCγ1 (Tyr783), rabbit polyclonal | Cell Signaling Technology | Cat# 2821S RRID:AB_330855 | WB (1:1000) |
| Antibody | GAPDH (14C10) Rabbit monoclonal | Cell Signaling Technology | Cat# 2118S RRID:AB_561053 | WB (1:1000) |
| Antibody | Anti-rabbit IgG, HRP-linked Antibody (Goat anti-rabbit) | Cell Signaling Technology | Cat# 7074S RRID:AB_2099233 | WB (1:1000) |
| Antibody | fluorescence-conjugated secondary antibodies (Donkey anti-rabbit 647) | Jackson ImmunoResearch | Cat# 711-605-152 RRID:AB_2492288 | WB (1:1000) |
| Recombinant DNA reagent | pACU2 (plasmid) | PMID:21606367 | RRID:Addgene_31223 | |
| Chemical compound, drug | VECTASHIELD Antifade Mounting Medium (with DAPI) | Vector Laboratories | Cat# H-1200 | |
| Software, algorithm | ImageJ (Fiji) | http://fiji.sc | RRID:SCR_002285 | |
| Software, algorithm | NeuronStudio | https://biii.eu/neuronstudio | RRID:SCR_013798 | |

## Cell lines

HEK293T cells were provided by Dr. Linfeng Chen, BHK21 cells were provided by Dr. Xianlin Nan, PC12 cells were a gift from Dr. Tobias Meyer. Confirmation of cell line authentication was done in the Cancer Center at Illinois at the University of Illinois at Urbana-Champaign. Mycoplasma contamination was done by a PCR-based protocol. All cell lines were kept at low passages in order to maintain their health and identity. Cell lines used in this work are not among the commonly misidentified cell lines maintained by the International Cell Line Authentication Committee.

## Molecular cloning

The plasmid of FOXO3-EGFP generated by cloning the human FOXO3 gene-containing plasmid (a gift from Prof. Anne Brunet at Stanford University) into the pEGFP-N1 backbone using overlap PCR with the (forward primer: cggactcagatctcgacgccaccatgtacccatacgatgttccggattacgc and the reverse primer: ccatggtggcgaccggtggatccccctgcttagcaccagt).

## Fly stocks

19–12-*Gal4* (*Xiang et al., 2010*), *repo-Gal80* (*Awasaki et al., 2008*), *ppk-CD4-tdGFP* (*Han et al., 2011*), and *ppk-Gal4 Han et al., 2011* have been previously described. To generate the *UAS-opto-Raf and UAS-optoAKT* stocks, we cloned the entire coding sequences into the pACU2 vector, and the constructs were then injected (Rainbow Transgenic Flies, Inc). Randomly selected male and

female larvae were used. Analyses were not performed blind to the conditions of the experiments. The experimental procedures have been approved by the Institutional Biosafety Committee (IBC) at the Children's Hospital of Philadelphia.

### Sensory axon lesion in *Drosophila*

Da neuron axon lesion and imaging in the PNS was performed in live fly larvae as previously described (*Song et al., 2012*; *Stone et al., 2014*; *Song et al., 2015*). VNC injury was performed as previously described (*Song et al., 2012*; *Li et al., 2020*). In brief, A3 and A6 axon bundles in the VNC were ablated with a focused 930 nm two-photon laser and full degeneration around the commissure junction was confirmed 24 hr AI. At 48 hr AI, axon regeneration of these two commissure segments were assayed independently of each other (*Figure 5—figure supplement 1*).

### Quantitative analyses of sensory axon regeneration in flies

Quantification was performed as previously described (*Song et al., 2012*; *Song et al., 2015*). Briefly, for axon regeneration in the PNS, we used 'regeneration percentage', which depicts the percent of regenerating axons among all the axons that were lesioned; 'regeneration index', which was calculated as an increase of 'axon length'/'distance between the cell body and the axon converging point (DCAC)' (*Figure 3—figure supplement 1A and B*). An axon was defined as regenerating only when it obviously regenerated beyond the retracted axon stem, and this was independently assessed of the other parameters. The regeneration parameters from various genotypes were compared with that of the WT if not noted otherwise, and only those with significant differences were labeled with the asterisks. For VNC injury, the increased length of each axon regrowing beyond the lesion sites was measured and added together. The regeneration index was calculated by dividing the sum by the distance between A4 and A5 axon bundles (*Figure 5—figure supplement 1*). Regeneration percentage was assessed independently of the regeneration index. A commissure segment was defined as regenerated only when at least one regenerating axon passed the midline of the commissure region or joined into other intact bundles (*Figure 5—figure supplement 1*).

### Live imaging in flies

Live imaging was performed as described (*Emoto et al., 2006*; *Parrish et al., 2007*). Embryos were collected for 2–24 hr on yeasted grape juice agar plates and were aged at 25°C or room temperature. At the appropriate time, a single larva was mounted in 90% glycerol under coverslips sealed with grease, imaged using a Zeiss LSM 880 microscope, and returned to grape juice agar plates between imaging sessions.

### Behavioral assay

The behavioral test was performed to detect functional recovery after VNC injury as described (*Li et al., 2020*). A7 and A8 C4da neuron axon bundles in the VNC, which correspond to the A7 and A8 body segments in the periphery, were injured with laser (*Figure 6A*). Since C4da neurons are essential for thermonociception, such lesion results in impaired nociceptive response to noxious heat at body segments A7 and A8. We assessed larva nociceptive behavior in response to a 47°C heat probe at 24 and 48 hr AI. At each time point, the larva was subjected to three consecutive trials, separated by 15 s (s). In each trial, the heat probe was applied at the A7 and A8 body segments for 5 s. If the larva produced head rolling behavior for more than two cycles, it would be scored as '1', otherwise '0' (*Figure 6A*). The scores of the three trials were combined and the total score at 24 hr AI was used to determine whether A7 and A8 bundles were successfully ablated. A larva was defined as recovered only when its total score was below 1 at 24 hr AI but increased to 2 or 3 at 48 hr AI. Those failed to exhibit such improvement at 48 hr AI were defined as unrecovered. All the injured larvae exhibited normal nociceptive responses when the same heat probe was applied at the A4 or A5 body segment at 24 hr AI.

### Immunohistochemistry

Third instar larvae or cultured neurons were fixed according to standard protocols. The following antibodies were used: rabbit anti-Phospho-p44/42 MAPK (Erk1/2) (Thr202/Tyr204) (4370, 1:100, Cell Signaling), rabbit anti-Phospho-*Drosophila* p70 S6 Kinase (Thr398) (9209S, 1:400, Cell Signaling) and

fluorescence-conjugated secondary antibodies (1:1000, Jackson ImmunoResearch). Larval body walls were mounted in VECTASHIELD Antifade Mounting Medium.

## Cell culture and transfection

HEK293T cells were cultured in DMEM medium supplemented with 10% fetal bovine serum (FBS), and $1 \times$ Penicillin Streptomycin solution (complete medium). Cultures were maintained in a standard humidified incubator at 37°C with 5% $CO_2$. For western blots, 800 ng of DNA were combined with 2.4 µL Turbofect in 80 µL of serum-free DMEM. The transfection mixtures were incubated at room temperature for 20 min prior to adding to cells cultured in 35 mm dishes with 2 mL complete medium. The transfection medium was replaced with 2 mL serum-free DMEM supplemented with $1 \times$ Penicillin Streptomycin solution after 3 hr of transfection to starve cells overnight. PC12 cells were cultured in F12K medium supplemented with 15% horse serum, 2.5% FBS, and $1 \times$ Penicillin Streptomycin solution. For PC12 neuritogenesis assays, 2400 ng of DNA were combined with 7.2 mL of Turbofect in 240 mL of serum-free F12K. The transfection medium was replaced with 2 mL complete medium after 3 hr of transfection to recover cells overnight. Twenty-four hours after recovery in high-serum F12K medium (15% horse serum + 2.5% FBS), the cell culture was exchanged to a low-serum medium (1.5% horse serum + 0.25% FBS) to minimize the base-level ERK activation induced by serum.

## Optogenetic stimulation for cell culture

For western blot analysis, transfected and serum-starved cells were illuminated for different time using a home-built blue LED light box emitting at 0.5 mW/cm$^2$. For PC12 cell neuritogenesis assay, PC12 cells were illuminated at 0.2 mW/cm$^2$ for 24 hr with the light box placed in the incubator.

## Optogenetic stimulation for fly

The whole optogenetics setup is modified from previous work (*Kaneko et al., 2017*). Larvae were grown in regular brown food at 25°C in 12 hr-12 hr light-dark cycle. At 72 hr AEL, early 3rd instar larvae were transferred from food, anesthetized with ether for axotomy. After recovery in regular grape juice agar plates, larvae were kept in the dark or under blue light stimulation thereafter. A 470 nm blue LED (LUXEON Rebel LED) was set over the grape-agar plate for stimulation. The LED was mounted on a 10 mm square coolbase and 50 mm square ×25 mm high alpha heat sink and set under circular beam optic with integrated legs for parallel even light. The light pattern was programmed with BASIC Stamp 2.0 microcontroller and buckpuck DC driver (LUXEON, 700 mA, externally dimmable).

Local light stimulation was delivered by a 488 nm argon-ion laser using a Zeiss LSM 880 microscope. At 24 hr AI, larvae were anesthetized and C4da neurons were imaged with a confocal microscope. For lesioned axons that bifurcated and formed two branches, we focused the laser beam (at 15% laser power) on the ventral branch for 5 min. The larva was then returned to grape juice agar plates and imaged again at 48 hr AI to assess the increased length of each branch.

## Live cell imaging

For the light-induced membrane recruitment assay, BHK-21 cells were co-transfected with optoRaf or optoAKT. Fluorescence imaging of the transfected cells was carried out using a confocal microscope (Zeiss LSM 700). GFP fluorescence was excited by a 488 nm laser beam; mCherry fluorescence was excited by a 555 nm laser beam. Excitation beams were focused via a 40 × oil objective (Plan-Neofluar NA 1.30). Ten pulsed 488 nm and 555 nm excitation were applied for each membrane recruitment experiment. CRY2-CIBN binding induced by 488 nm light was monitored by membrane recruitment of CRY2-mCherry-Raf1 (for optoRaf) or CRY2-mCherry-AKT (for optoAKT) to the CIBN-CIBN-GFP-CaaX-anchored plasma membrane. The powers after the objective for 488 nm and 555 nm laser beam are approximately 40 µW and 75 µW, respectively. Alternatively, an epi-illumination fluorescence microscope (Leica DMI8) equipped with a 100 × objective (HCX PL FLUOTAR 100×/1.30 oil) and a light-emitting diode illuminator (SOLA SE II 365) was used for the CRY2-mCherry-Raf1 membrane translocation assay. Neurite outgrowth of PC12 cells was imaged using an epi-illumination fluorescence microscope (Leica DMI8) equipped with 10× (PLAN 10×/0.25) and 40× (HCXPL FL L 40×/0.6) objectives. Fluorescence from GFP was detected using the GFP filter cube (Leica,

excitation filter 472/30, dichroic mirror 495, and emission filter 520/35); fluorescence from mCherry was detected using the Texas Red filter cube (Leica, excitation filter 560/40, dichroic mirror 595, and emission filter 645/75).

## Western blot

Cells were washed once with 1 mL cold DPBS and lysed with 100 µL cold lysis buffer (RIPA + protease/phosphatase cocktail). Lysates were centrifuged at 17,000 RCF, 4°C for 10 min to pellet cell debris. Purified lysates were normalized using Bradford reagent. Normalized samples were mixed with LDS buffer and loaded onto 10% or 12% polyacrylamide gels. SDS-PAGE was performed at room temperature with a cold water bath. Samples were transferred to PVDF membranes at 30 V 4°C overnight or 80 V for 90 min. Membranes were blocked in 5% BSA/TBST for 1 hr at room temperature and probed with the primary and secondary antibodies according to company guidelines. Membranes were incubated with ECL substrate and imaged using a Bio-Rad ChemiDoc XRS chemiluminescence detector. Signal intensity analysis was performed by ImageJ. Antibodies used were listed in the Key Resources Table.

## Statistical analysis

No statistical methods were used to pre-determine sample sizes but our sample sizes are similar to those reported in previous publications (*Song et al., 2012*; *Song et al., 2015*), and the statistical analyses were done afterward without interim data analysis. Data distribution was assumed to be normal but this was not formally tested. All data were collected and processed randomly. Each experiment was successfully reproduced at least three times and was performed on different days. The values of '*N*' (sample size) are provided in the figure legends. Data are expressed as mean ± SEM in bar graphs, if not mentioned otherwise. No data points were excluded. Two-tailed unpaired Student's t-test was performed for comparison between two groups of samples. One-way ANOVA followed by multiple comparison test was performed for comparisons among three or more groups of samples. Two-way ANOVA followed by multiple comparison test was performed for comparisons between two or more curves. Fisher's exact test was used to compare the percentage. Statistical significance was assigned, *p<0.05, **p<0.01, ***p<0.001.

## Acknowledgements

This work was supported by NIH grants 1R01NS107392 (YS) and 1R01GM132438 (KZ).

## Additional information

### Funding

| Funder | Grant reference number | Author |
| --- | --- | --- |
| National Institute of General Medical Sciences | R01GM132438 | Huaxun Fan<br>Savanna S Skeeters<br>Vishnu V Krishnamurthy<br>Kai Zhang |
| National Institute of Neurological Disorders and Stroke | 1R01NS107392 | Qin Wang<br>Feng Li<br>Yuanquan Song |

The funders had no role in study design, data collection and interpretation, or the decision to submit the work for publication.

### Author contributions

Qin Wang, Data curation, Formal analysis, Writing - original draft, Writing - review and editing; Huaxun Fan, Data curation, Formal analysis, Methodology, Writing - original draft; Feng Li, Savanna S Skeeters, Vishnu V Krishnamurthy, Data curation, Formal analysis; Yuanquan Song, Kai Zhang, Conceptualization, Supervision, Funding acquisition, Investigation, Methodology, Writing - original draft, Project administration, Writing - review and editing

## Author ORCIDs
Vishnu V Krishnamurthy http://orcid.org/0000-0001-9905-5965
Yuanquan Song https://orcid.org/0000-0001-7699-2059
Kai Zhang https://orcid.org/0000-0002-6687-4558

## Ethics
Animal experimentation: The experimental procedures have been approved by the Institutional Biosafety Committee (IBC) at the Children's Hospital of Philadelphia.

## Decision letter and Author response
Decision letter https://doi.org/10.7554/eLife.57395.sa1
Author response https://doi.org/10.7554/eLife.57395.sa2

# Additional files

## Supplementary files
• Transparent reporting form

## Data availability
All data generated or analysed during this study are included in the manuscript and supporting files.

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
