## [Decision Letter]

**Acceptance summary:**

This study cleverly uses ontogenetic probes to stimulate the Raf and Akt pathways in the *Drosophila* peripheral and central nervous systems. Using light-activated signaling allows for the dissection and regulation of dendritic branching and axon regeneration. The manuscript has been strengthened with additional controls and experiments to increase the specificity of Raf and Akt signaling through analysis of mutant versions of these enzymes. Both genetic and biochemical data are employed to support the conclusions of the study.

**Decision letter after peer review:**

Thank you for submitting your article "Optical control of ERK and AKT signaling promotes axon regeneration and functional recovery of PNS and CNS in *Drosophila*" for consideration by *eLife*. Your article has been reviewed by three peer reviewers, one of whom is a member of our Board of Reviewing Editors, and the evaluation has been overseen by Jonathan Cooper as the Senior Editor. The following individual involved in review of your submission has agreed to reveal their identity: Philip JS Stork (Reviewer #2).

The reviewers have discussed the reviews with one another and the Reviewing Editor has drafted this decision to help you prepare a revised submission.

This study uses ontogenetic probes to activate the Raf and Akt pathways. This approach represents a powerful system to dissect signaling outcomes of these pivotal enzymes. Light-controlled activation of these two relevant signals is used to assess their ability to mediate axonal pathfinding, branching, and regeneration. The optogenetic tools allow for an examination of the sufficiency of either Akt or ERK pathway activation in these processes, in both PNS and CNS neurons. The result that both pathways play significant yet distinct roles in these processes is important and is of wide interest.

Summary:

Several major and minor reservations were brought up by the reviewers. The major issues are to define more fully the upstream events that activate Raf-1 and Akt and the need for additional controls for the optogenetic probes. The kinetics of activation, as well as cell localization, requires attention. Another request is to examine if there is convergence in the two pathways. The major concerns are described below--

Essential revisions:

1) This study assumes that the tools trigger signaling pathways independently of upstream (neurotropic) signaling. However, whether these tools require some upstream signaling remain incompletely addressed. For example, activation of Raf1 requires upstream activation by kinases phosphorylating the N-terminal region (Y341 and S338). The phosphorylation of S338 is a commonly used read-out for Raf-1 activation (and mutants at this position show no activation). It would be very informative to examine the status of pS338 in optoRaf and to compare the optoRaf to a mutant S338A version, at least in Hek293 cells. Because these phosphorylations are linked to Raf dimerization, these studies would provide insight into whether Raf dimerization is required or possible in this context.

2) It would be also helpful to include more specificity controls for Raf vs. Akt signaling in *Drosophila* neurons to ensure the signals directly go to cells where the functional assessments are being conducted.

3) The kinetic experiments are interesting but somewhat incomplete, and it is unclear what the takeaway from these experiments should be. Importantly, it is not known how different pulsed light patterns translate temporally to signaling. It seems that from the data in Figure 1, it is possible that in neurons patterns may maintain a constant activation of the pathway. Additional controls looking at the extent of signaling in neurons with these paradigms would be really helpful.

---

## [Author Response]

Essential revisions:1) This study assumes that the tools trigger signaling pathways independently of upstream (neurotropic) signaling. However, whether these tools require some upstream signaling remain incompletely addressed. For example, activation of Raf1 requires upstream activation by kinases phosphorylating the N-terminal region (Y341 and S338). The phosphorylation of S338 is a commonly used read-out for Raf-1 activation (and mutants at this position show no activation). It would be very informative to examine the status of pS338 in optoRaf and to compare the optoRaf to a mutant S338A version, at least in Hek293 cells. Because these phosphorylations are linked to Raf dimerization, these studies would provide insight into whether Raf dimerization is required or possible in this context.

This is an excellent point. As suggested, we have performed Western blot in HEK293T cells using an antibody against phosphor-S338 of Raf1 under the optogenetic stimulation conditions. As a positive control, we used serum stimulation. As expected, in cells transfected with the optoRaf S338A mutant, optoRaf S338A cannot be phosphorylated at this site regardless of serum and light stimulation (Figure 1—figure supplement 1E, top panel). Importantly, unlike wild-type optoRaf, optoRaf S338A significantly abolished the light-dependent activation of ERK (Figure 1—figure supplement 1E, third panel). This result indicates that activation of optoRaf should require the upstream activation machinery. Furthermore, as suggested by the reviewer, it is very likely that Raf dimerization is involved in the activation of optoRaf.

2) It would be also helpful to include more specificity controls for Raf vs. Akt signaling in *Drosophila* neurons to ensure the signals directly go to cells where the functional assessments are being conducted.

Thanks for the suggestion. To make sure that light-mediated enhancement of Raf and AKT signaling occurs specifically in the target cell, we used DAPI staining to identify da neurons with or without the corresponding optogenetic systems. As shown in Figure 2—figure supplement 2, within the field of view, only C4da neurons marked by ppk-CD4tdGFP express optoRaf or optoAKT, which is under the control of ppk-Gal4, an enhancer driver for C4da neurons. In these cells, blue light-stimulation significantly enhances the level of phospho-ERK or phospho-p70 ribosomal S6 kinase (phospho-p70^S6K^) compared with the neighboring cells, which do not express optoRaf or optoAKT (DAPI only, no GFP). This result confirms that blue light specifically triggers Raf/MEK/ERK or AKT signaling in optoRaf/AKT expressing C4da neurons.

3) The kinetic experiments are interesting but somewhat incomplete, and it is unclear what the takeaway from these experiments should be. Importantly, it is not known how different pulsed light patterns translate temporally to signaling. It seems that from the data in Figure 1, it is possible that in neurons patterns may maintain a constant activation of the pathway. Additional controls looking at the extent of signaling in neurons with these paradigms would be really helpful.

We really appreciate the insightful comments. As suggested, we carried out a kinetic study in C4da neurons to determine if intermittent light pattern leads to intermittent signaling activity. The level of ERK and AKT signaling activities was probed by immunostaining of phospho-ERK and phospho-p70^S6K^, respectively. Consistent with the results from mammalian cell culture, 5-10 min of light stimulation was sufficient to activate optoRaf and optoAKT (Figure 2—figure supplement 1). After the light was off, the level of phospho-ERK and phospho-p70^S6K^ decreases monotonically. We observed that optoAKT inactivates faster than optoRaf, i.e., level of phospho-p70^S6K^ decreases to the basal level within 15 min, whereas phospho-ERK decreases slower (Figure 2). This difference in decay kinetics may arise from distinct signaling threshold, signaling capacity of Raf and AKT, or both.